# A chromatinized origin reduces the mobility of ORC and MCM through interactions and spatial constraint

Humberto Sánchez [1], Zhaowei Liu [1], Edo van Veen [1], Theo van Laar [1], John F. X. Diffley [2] & Nynke H. Dekker [1] ✉

Chromatin replication involves the assembly and activity of the replisome within the nucleosomal landscape. At the core of the replisome is the Mcm2-7 complex (MCM), which is loaded onto DNA after binding to the Origin Recognition Complex (ORC). In yeast, ORC is a dynamic protein that diffuses rapidly along DNA, unless halted by origin recognition sequences. However, less is known about the dynamics of ORC proteins in the presence of nucleosomes and attendant consequences for MCM loading. To address this, we harnessed an in vitro single-molecule approach to interrogate a chromatinized origin of replication. We find that ORC binds the origin of replication with similar efficiency independently of whether the origin is chromatinized, despite ORC mobility being reduced by the presence of nucleosomes. Recruitment of MCM also proceeds efficiently on a chromatinized origin, but subsequent movement of MCM away from the origin is severely constrained. These findings suggest that chromatinized origins in yeast are essential for the local retention of MCM, which may facilitate subsequent assembly of the replisome.

Eukaryotic genomes are structurally organized as a combination of DNA and proteins known as chromatin. Cell biology, genetics, and biochemical experiments have demonstrated that the quantity and local density of the basic structural unit of chromatin, the nucleosome, can define where the initiation of DNA replication happens and that chromatin folding influences replication timing[1]. Less is known about the influence of nucleosomes on the individual proteins that are involved in establishing DNA replication, including their dynamics.

DNA replication in eukaryotes occurs during the S phase of the cell cycle, during which the entire genome must be duplicated in a coordinated manner. In the budding yeast *Saccharomyces cerevisiae*, replication is initiated from hundreds of sequence-specific origins of replication. Each origin is prepared for replication initiation in two temporally separated steps that are essential for proper DNA replication[2]. During the G1 phase of the cell cycle, the Origin Recognition Complex (ORC) first binds to replication origins[3]. These origins

are autonomous replicating sequences (ARS) that contain both a strong binding site (the ARS consensus sequence, or ACS) and a weak binding site (B2) for ORC. Once bound, ORC, together with Cdc6 and Cdt1, then loads two inactive Mcm2-7 (MCM) replicative DNA helicases, forming the pre-replication complex (pre-RC); this is known as the licensing step[4,5]. During S phase, origins of replication are activated by the combined action of two kinases, S-CDK and DDK, and several other proteins known as firing factors, which serve to bring Cdc45 and GINS into association with MCM and form the CMG holo-helicase[6–8]. The addition of DNA polymerases as well as accessory proteins completes the assembly of the full replisome and allows DNA replication to proceed[8,9].

During the licensing step, ORC, Cdc6, and MCM/Cdt1 load at DNA replication origins that are embedded in the chromatin[2]. Biochemical studies have shown that such origins are free of nucleosomes[10], and genome-wide analysis suggests that these specific nucleosome-free

---

[1]Department of Bionanoscience, Kavli Institute of Nanoscience, Delft University of Technology, Delft, The Netherlands. [2]Chromosome Replication Laboratory, Francis Crick Institute, London, United Kingdom. ✉e-mail: n.h.dekker@tudelft.nl

regions favor ORC binding[11]. Interestingly, ORC then plays an active role in positioning nucleosomes around the origin[11,12] by coordinating the activity of chromatin remodelers before and during replication[12,13]. Chromatin remodelers that act at the origin also influence origin licensing[14].

We have previously demonstrated that yeast ORC is a mobile protein that rapidly diffuses on bare DNA, but that origin recognition halts this search process[15]. Furthermore, it has been observed that single and double MCM hexamers diffuse on DNA[4,5] and that nucleosomes can act as potential obstacles for MCM diffusion[16], so we sought to investigate the roles of origin-flanking nucleosomes in either directly limiting or locally targeting the diffusion of ORC and, consequently, in the loading of MCM onto DNA.

## Results

### Establishment of a labeled chromatinized origin of replication within 10.4 kbp DNA for single-molecule investigations

To experimentally examine the origin localization and dynamics of ORC and MCM on a chromatinized origin of replication, we designed an in vitro single-molecule force-fluorescence assay using purified yeast proteins (Supplementary Fig. 1.1). We wished to visualize the chromatinized origin using yeast histone octamers marked with fluorescent labels. To achieve this, we first introduced a single cysteine on the H2A histone by replacing residue lysine 120 (K120), which avoids disruption of the overall nucleosome structure (Fig. 1a; Methods). Purified yeast histone octamers containing two H2A (K120C) histones were then covalently bound with one single AF488 dye per cysteine (degree of labeling $p_{label} = 0.81$, as determined from bulk experiments; Methods), hereafter referred to as H2A$^{AF488}$. We next engineered an ARS1 origin of replication flanked by two nucleosome positioning sequence (NPS) sites spaced by 144 bp (~50 nm) (Supplementary Fig. 1.2a, ref. 16, and Methods), and reconstituted fluorescently labeled nucleosomes onto these NPS sites using salt gradient dialysis (Supplementary Fig. 1.2b). The resulting chromatinized origin

tested positively for *Pst*I restriction digestion between the nucleosomes (Supplementary Fig. 1.2c), which indicated that it should remain accessible to ORC and MCM. Next, we ligated this chromatinized origin to two biotinylated DNA fragments of distinct sizes. This resulted in a 10.4 kilobase pair (kbp) chromatinized DNA molecule (Fig. 1b) in which the origin was localized at approximately one-third of the total length of the DNA molecule (Supplementary Fig. 1.2d). In addition to the ARS1 origin, the 10.4 kbp DNA molecule contained a number of endogenous potential binding sites for ORC[17] and regions of high AT content (Supplementary Fig. 1.2e).

Following preparation, the 10.4 kbp DNA molecules including a chromatinized origin were introduced into the microfluidic flow cell of our single-molecule instrument. Here, the DNA molecules were tethered to streptavidin-coated beads in a dual-beam optical trap, after which they were visualized in a protein-free channel of the flow cell (buffer channel) while under a tension of 2 pN. Nucleosome-bound H2A$^{AF488}$ was detected as a bright fluorescent spot (or focus) whose spatial position was determined (Fig. 1b). By repeating this measurement on multiple DNA molecules, we could build up the spatial distribution of H2A$^{AF488}$ foci as a function of genomic coordinate. As each individual DNA molecule was tethered in one of two possible orientations in the dual-beam optical trap, we report spatial distributions from the midpoint of the DNA. Each histogram bin (width 0.670 kbp; Methods) thus contains the average of the occupancies of two segments of DNA located symmetrically about the midpoint of the DNA: for example, the origin bin contains the mean of the occupancies of the actual origin- and NPS-containing segment and the DNA segment located oppositely from the midpoint of the DNA by 1.7 kbp. The spatial distribution of H2A$^{AF488}$ foci (filtered to remove foci containing more than four H2A$^{AF488}$, which could represent aggregates; Methods) showed a clear peak in the bin containing the NPS sites (left panel in Fig. 1c), suggesting preferential nucleosome assembly at the NPS sites. This spatial distribution of H2A$^{AF488}$ was built up from distinct DNA samples chromatinized on different days, which exhibited only

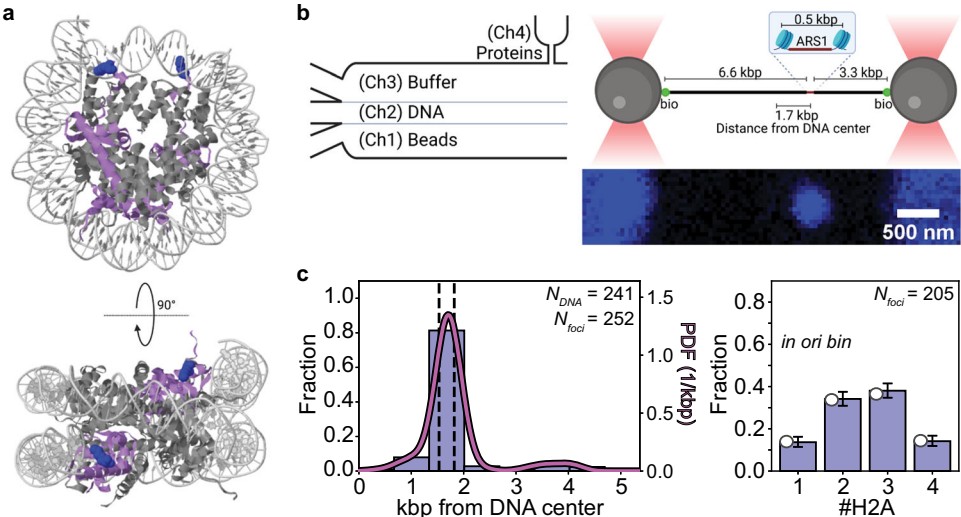

**Fig. 1 | Design and characterization of a labeled chromatinized origin within 10.4 kbp DNA. a** Structural depictions of the yeast nucleosome (PDB:1ID3), including in blue the mutated residue on the H2A histones for adding fluorescent labels. **b** (left panel) Schematic of the flow cell used in the single-molecule experiments. (right upper panel) Schematic of the 10.4 kilobase pair (kbp) DNA held in an optical trap that contains the ARS1 origin of replication flanked by two nucleosome positioning sites (NPSs). The DNA is chromatinized via salt gradient dialysis prior to its introduction into the single-molecule flow cell (Supplementary Fig. 1.2). Created with BioRender.com. (right lower panel) Confocal scan showing that signal from H2A$^{AF488}$ is detected as a single diffraction-limited spot localized at the NPSs. **c** (left panel) Spatial distribution of H2A$^{AF488}$ on DNA described in panel

(b), acquired immediately after introduction into the flow cell and deduced from the blue diffraction-limited spots ($N_{foci}$) collected from 248 distinct DNA molecules ($N_{DNA}$). Dashed lines indicate the location of the NPSs, and the solid curve indicates the kernel density estimation of the data (PDF: probability density function). (right panel) Stoichiometry distribution of H2A$^{AF488}$ in the bin containing the chromatinized origin. Data are presented as mean values ± one-sigma Wilson confidence intervals. Filled white circles indicated at left designate the fitted values based on the model described in the Methods and in Supplementary Fig. 1.3. Data in both panels derives from four chromatinized samples (Supplementary Fig. 1.4). Source data are provided as a Source Data file.

minimal differences between them (Supplementary Fig. 1.4), highlighting the reproducibility of our sample preparation.

Ideally, our chromatinized origin should contain two fluorescently labeled nucleosomes within a single diffraction-limited spot. Assuming that the identification of a single H2A[AF488] reflects the presence of a single H2A[AF488]-H2B dimer bound to a H3–H4 tetrasome, we should then measure 4 H2A[AF488] per focus. However, the experimental counts of H2A[AF488] may be reduced by non-unity values for the probability of NPS site occupancy (by tetrasome, hexasome, or full nucleosome; $p_{occupancy}$), the probability of H2A[AF488]-H2B dimer association with the H3-H4 tetrasome ($p_{h2a}$), and $p_{label}$ (Methods, Supplementary Fig. 1.3a). To count the number of H2A[AF488], we identified step-wise photobleaching events using Change-Point Analysis (CPA; Methods), whereby the methodology is assessed and validated using dCas9[AF488] (Supplementary Fig. 1.1). The lifetime prior to photobleaching of AF488 when linked to H2A was similar to that measured when it was linked to dCas9, indicating that H2A[AF488] histones are stable on the DNA during the experiment (Supplementary Fig. 1.3c). Using this photobleaching approach, we found that the H2A[AF488] foci primarily contained two to three H2A[AF488], as shown in the stoichiometry distribution (right panel in Fig. 1c) built up from a total of $N_{DNA}$ = 247 derived from the four different DNA samples mentioned above (again exhibiting only minimal differences between them, Supplementary Fig. 1.4). With the independently measured $p_{label}$ = 0.81 as fixed parameter, the best fit to the experimentally determined stoichiometry distribution (Fig. 1c, with mean squared error (MSE) = $5.18 \times 10^{-5}$ and fit values indicated as filled white circles at left on the bars; Supplementary Fig. 1.3b) yielded $p_{occupancy}$ = 1.00 (corresponding to full occupancy of both NPS sites) and $p_{h2a}$ = 0.77.

To confirm that the detected H2A[AF488] foci reflected embedding of H2A[AF488] histones into nucleosomes—as opposed to unspecific electrostatic interactions of H2A[AF488]-H2B dimers with the DNA[18]—we separately probed for nucleosome presence on our 10.4 kbp DNA using force spectroscopy. Previous experiments have shown that under force, DNA can be unwrapped from the histone octamer. Irreversible jumps in the force–extension curve in which 27 nm (80 bp) of DNA is unwrapped from either a hexasome or a full nucleosome have been reported to occur over a broad range of forces (8–40 pN)[19–22]. Indeed, when we pulled on one extremity of a tethered chromatinized DNA molecule (formed with H2A[AF488]) at a constant speed (100 nm/s) (Supplementary Fig. 1.5a), we could identify such jumps (Supplementary Fig. 1.5b). These occurred at a force of 17.6 ± 5.8 pN (mean ± standard deviation) and yielded contour length increments of 24.8 ± 5.9 nm (mean ± standard deviation), the latter corresponding to the contour length of the unwrapped DNA. At least two such jumps were revealed in 88.5% of the chromatinized DNA molecules probed (Supplementary Fig. 1.5c), implying that our chromatinized DNA typically contains two nucleosomes (or hexasomes). Quantitatively, this agrees well with the percentage of H2A[AF488] foci containing two or more H2A[AF488] of 96% computed from the fit parameters $p_{occupancy}$ = 1.00 and $p_{h2a}$ = 0.77 of the fluorescence data described above.

Together, these single-molecule fluorescence and force spectroscopy results support our establishment of a 10.4 kbp DNA containing a single origin of replication flanked by two nucleosomes. As the force spectroscopy does not report on nucleosome location, however, to investigate the influence of a chromatinized origin of replication on ORC and MCM, we focused on fluorescence readouts alone.

## Nucleosomes enhance intrinsic ORC preference for the origin of replication

To study the effect of chromatinized origins on ORC binding and dynamics, we labeled the N-terminus of the Orc3 subunit with a JF646 fluorophore via a HaloTag (Methods), hereafter referred to as ORC[JF646]. We confirmed that the ORC[JF646] could load MCM in bulk assays

(Supplementary Fig. 1.1b). We then prepared four different 10.4 kbp DNA molecules that either contained ARS1 or a mutated origin without specific affinity for ORC[15,23], and that were chromatinized or not.

We first performed experiments that report on the rapid binding of ORC[JF646] to a chromatinized origin (formed with H2A[AF488], as above). To do so, we incubated the optically trapped DNA molecule held at near-zero force in a reservoir of the microfluidic flow cell containing 5 nM ORC[JF646] for 5 s. Subsequently, we shifted the DNA to a separate, protein-free channel and imaged it under a stretching force of 2 pN (Fig. 2a). DNA-bound ORC[JF646] could then be observed as a bright fluorescent spot (focus), after which the experiment was repeated with a new DNA. Stoichiometric analysis of the foci via step-wise photobleaching indicated that they predominantly contained individual ORC[JF646] molecules (Supplementary Fig. 2). Foci containing more than five ORC[JF646], which could represent aggregates (Methods), were not analyzed further. The remaining fluorescent foci were observed throughout the DNA molecule, but the overall spatial distribution exhibited clear overrepresentation of ORC[JF646] in the bin containing the ARS1 origin (30% of the total, Fig. 2b-i). This preference for ORC[JF646] binding in the bin containing the origin[15] increased to 39% when the origin was chromatinized (Fig. 2b-ii). Mutation of the origin reduced preferential binding by ORC[JF646] in the bin containing the origin[15]; instead, ORC[JF646] was observed to peak in an adjacent bin (Fig. 2b-iii) that included a potential ORC binding site (located at 2.4 kbp from the start of the construct, Supplementary Fig. 1.2d). Chromatinization of this mutated origin nonetheless increased ORC[JF646] presence in the origin bin from 14% to 26% (Fig. 2b-iii, 2b-iv). For the four conditions probed, ORC[JF646] binding in the bin containing the origin (chromatinized or not) was subjected to tests of statistical significance (Fig. 2c). These results indicate a statistically significant enhancement of ORC[JF646] binding in the presence of either ARS1 or nucleosomes, with the effect of the former being more pronounced.

## Nucleosomes flanking the origin reduce ORC mobility and increase ORC lifetime

We have previously shown that ORC is a mobile protein that slides on bare DNA[15]. Here we sought to investigate the motion dynamics of ORC on DNA containing a chromatinized origin following rapid binding under the incubation conditions described above (Fig. 2a; see sample kymographs in Supplementary Fig. 3.1). When we tracked the position of ORC[JF646] molecules initially located in the bin containing the chromatinized origin over time (Fig. 3a-i, showing 30% of all traces, selected at random), it appeared that within experimental error many of these ORC[JF646] molecules hardly changed their position. Conversely, ORC[JF646] molecules initially located outside the origin (Fig. 3a-ii, showing 30% of all traces, selected at random) were observed to explore their local DNA environment in seemingly random fashion, which in some cases allowed them to approach, and then move away from, the nucleosomes.

To quantify these motion dynamics of ORC, we calculated the mean squared displacement for each ORC molecule on the four types of DNA molecules described above versus time interval and extracted a diffusion constant from a linear fit. Distributions (in log scale) of the fitted diffusion constants are shown in Fig. 3b. We observed a wide spread of diffusion constants, ranging from $10^{-6}$ to $10^{0}$ kbp$^2$ s$^{-1}$. Using the Bayesian Information Criterion, we identified kinetically distinct population states for ORC on non-chromatinized DNA containing ARS1 (Fig. 3b-i). These describe either a population with a low mean diffusion constant (hereafter slow ORC population) ($D_{slow}$ = 0.0049 ± 0.0028 kbp$^2$ s$^{-1}$ (mean ± SEM)) or a population with a higher mean diffusion constant (hereafter fast ORC population) ($D_{fast}$ = 0.152 ± 0.023 kbp$^2$ s$^{-1}$), consistent with our earlier findings for ORC on longer DNA molecules that contained a synthetic origin of replication[15]. We used these means as imposed values in fitting the distribution of ORC diffusion constants obtained on the other three

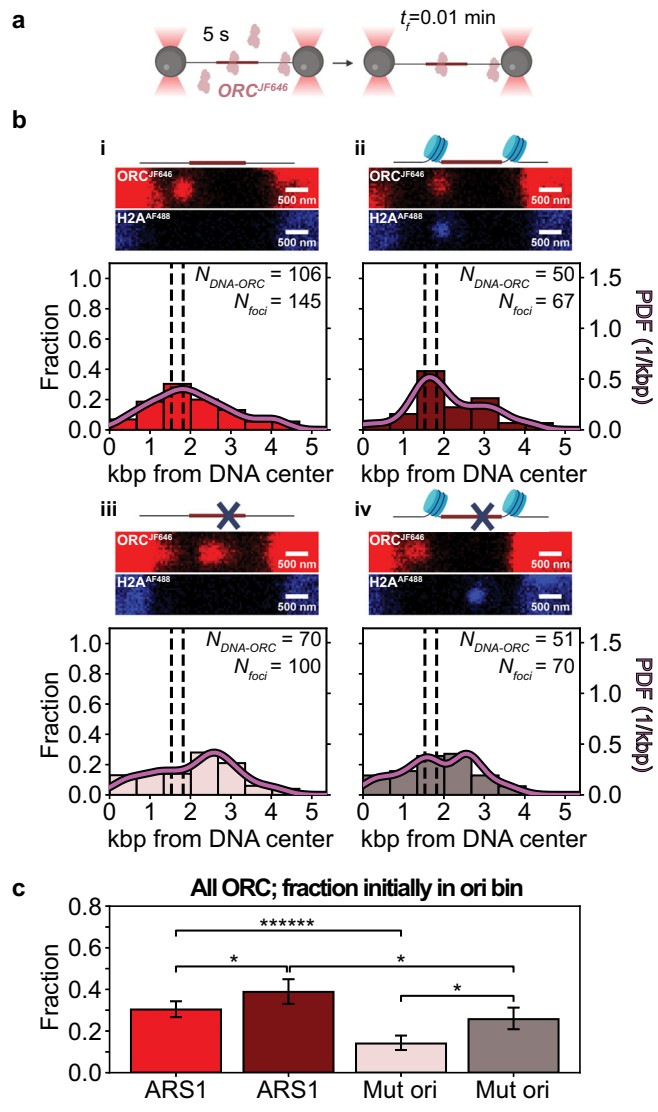

**Fig. 2 | Spatial distribution of rapidly bound ORC$^{JF646}$ on 10.4 kbp DNA containing an ARS1 (or mutated) origin (chromatinized or not). a** Tethered DNA molecules are introduced into the protein channel for a 5 s incubation with ORC$^{JF646}$ and Cdc6 in the presence of ATP, followed by confocal scanning in the buffer channel at an acquisition frequency of one frame every 0.01 min ($t_f$ = 0.01 min). Created with BioRender.com. **b** (i–iv) Schematics of the origin regions, representative fluorescence scans over the full length of the DNA (red: ORC$^{JF646}$; blue, H2A$^{AF488}$; note that DNA molecules can be captured in opposite orientations), and spatial distributions of ORC$^{JF646}$ acquired immediately after incubation in Ch4 and initial illumination in Ch3. Created with BioRender.com. Focusing on the latter: (i) Spatial distribution of ORC$^{JF646}$ on DNA containing a non-chromatinized origin as deduced from the red diffraction-limited spots ($N_{foci}$) collected from 106 distinct DNA molecules ($N_{DNA-ORC}$). The dashed lines indicate the location of the NPSs, and the solid curve indicates the kernel density estimation of the data (PDF: probability density function). (ii) Spatial distribution of ORC$^{JF646}$ on DNA containing a chromatinized origin as deduced from the red diffraction-limited spots ($N_{foci}$) collected from 50 distinct DNA molecules ($N_{DNA-ORC}$) analyzed and displayed as in (**b**-i). (iii) Spatial distribution of ORC$^{JF646}$ on DNA containing a non-chromatinized mutated origin as deduced from the diffraction-limited red spots ($N_{foci}$) collected from 70 distinct DNA molecules ($N_{DNA-ORC}$) analyzed and displayed as in (**b**-i). (iv) Spatial distribution of ORC$^{JF646}$ on DNA containing a chromatinized mutated origin as deduced from the red diffraction-limited spots ($N_{foci}$) collected from 51 distinct DNA molecules ($N_{DNA-ORC}$) analyzed and displayed as in (**b**-i). **c** ORC$^{JF646}$ occupancy probability for the bins containing the (chromatinized or not) origin in (**b**-i)–(**b**-iv), as indicated by the corresponding color bar (ARS1: non-mutated origin; Mut ori: mutated origin; nuc: chromatinized). $N_{foci-ARS1}$ = 145; $N_{foci-ARS1+nuc}$ = 67; $N_{foci-Mut\ ori}$ = 100; $N_{foci-Mut\ ori+nuc}$ = 70. Data are presented as mean values ± one-sigma Wilson confidence intervals. Statistical significance is determined by a two-sided binomial test ($p$ value$_{ARS1/ARS1+nuc}$ = 4.0 × 10$^{-2}$; $p$ value$_{ARS1/Mut\ ori}$ = 4.2 × 10$^{-9}$; $p$ value$_{ARS1+nuc/Mut\ ori+nuc}$ = 1.7 × 10$^{-2}$; p-value$_{Mut\ ori/Mut\ ori+nuc}$ = 1.3 × 10$^{-2}$): * $p$ < 0.05, ****** $p$ < 0.0000005. Source data are provided as a Source Data file.

origin also led to an increase in the slow population of ORC molecules, again irrespective of the origin contained ARS1 or was mutated (Fig. 3c-iii). This reduction in ORC mobility suggested ORC interactions with the nucleosomes.

To address whether chromatinization of the origin influenced the stability of ORC binding, we examined the lifetimes of individual JF646 dyes on DNA-bound ORC$^{JF646}$ molecules by tracking foci containing 1 or 2 ORC$^{JF646}$ molecules at 0.6 s/frame until the fluorescence signal disappeared. Apart from the DNA molecule with the chromatinized mutated origin, the mean lifetime of ORC$^{JF646}$ was shorter than 30 s, making it substantially shorter than the bleaching-limited mean lifetime of JF646 dyes measured under identical imaging conditions (71.0 s, Supplementary Fig. 1.1c) by attaching it to DNA-bound dCas9$^{JF646}$. This indicated that ORC$^{JF646}$ molecules typically dissociated from DNA during the measurement, in accordance with our previous observations[15]. Nonetheless, the mean lifetime of the slow ORC population typically exceeded that of the fast ORC population (Supplementary Fig. 3.2b). Of the ORC molecules initially located at or close to the ARS1 origin on bare DNA, 61% were associated with the slow ORC population (red bin, Fig. 3c-ii); this fraction is increased to 84% upon chromatinization of the ARS1 origin (bordeaux bin, Fig. 3c-ii). While not all of these ORC molecules may be specifically bound to DNA (as also suggested by a corresponding increase in the slow ORC population observed upon chromatinization of the mutated origin, compare gray and pink bins in Fig. 3c-ii), nonetheless, the increase in the slow ORC population with its higher mean lifetime on a DNA containing chromatinized ARS1 suggests that flanking nucleosomes can contribute to the temporal retention of ORC near an origin of replication. This could, in turn, contribute to preferential recruitment of MCM there.

## Stable ORC binding to DNA requires the origin of replication
We also performed experiments with more extended incubation conditions compatible with the timescale required for full maturation

types of DNA molecules (Fig. 3b-ii, 3b-iii, and 3b-iv). As one can observe both by eye and from the fits, chromatinization of the ARS1 origin resulted in an increase in the slow ORC population (Fig. 3b-ii). On DNA molecules with the mutated origin, the fast ORC population predominated (Fig. 3b-iii), again consistent with our previous findings[15] but chromatinization of this mutated origin also resulted in an increase in the slow ORC population (Fig. 3b-iv). We summarize these results more quantitatively by computing the fraction of the slow ORC population in the total, whereby we used the geometric mean $D^*$ ($=10^{(\mu_{slow} + \mu_{fast})/2}$ = 0.0065 kbp$^2$ s$^{-1}$, where $\mu_i$ are the lognormal population means) as a cutoff (Fig. 3c-i). This fraction was always observed to be higher upon chromatinization of the origin.

To assess how chromatinization of the origin resulted in an increase of the slow ORC population, we repeated this analysis separately for ORC molecules initially localized in the origin bin (Fig. 3c-ii) and for ORC molecules initially localized elsewhere (Fig. 3c-iii). For ORC molecules initially localized in the origin bin, chromatinization of the origin led to an increase in the slow ORC population, irrespective of whether the origin contained ARS1 or was mutated (Fig. 3c-ii). This reduction in ORC mobility could have resulted solely from the spatial confinement imposed on ORC by the nucleosomes, or additionally through ORC interactions with the nucleosomes. Interestingly, for ORC molecules initially localized elsewhere, chromatinization of the

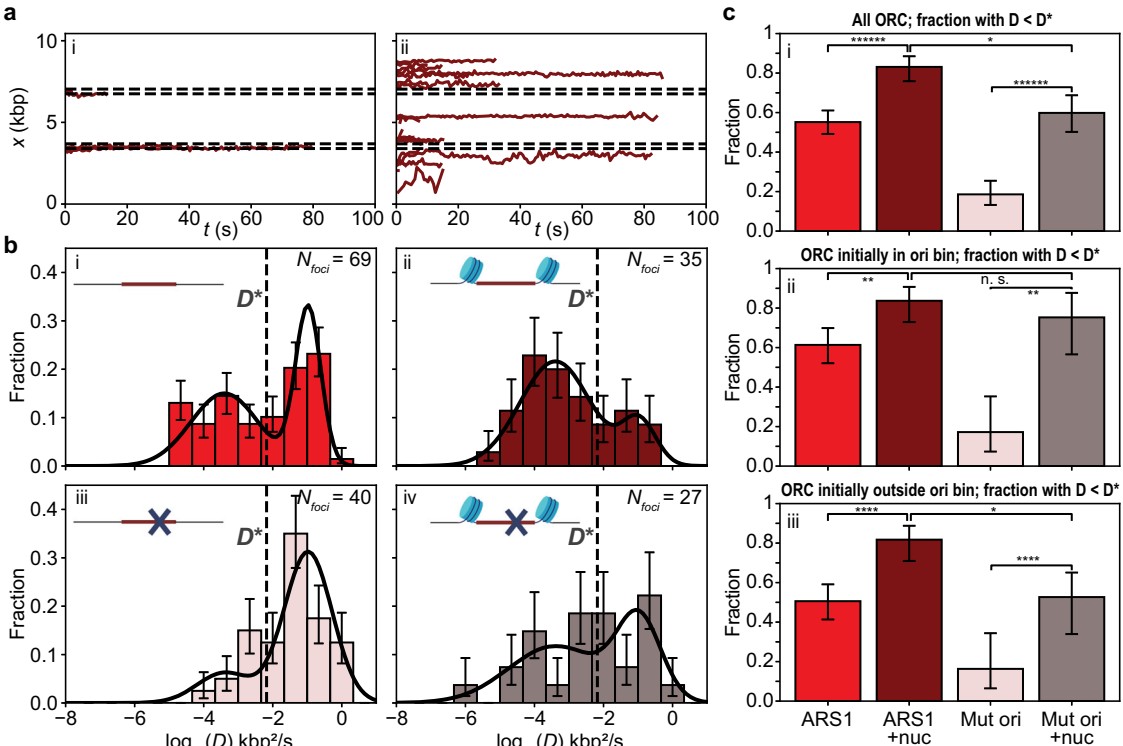

**Fig. 3 | Mobility of rapidly bound ORC^JF646 on 10.4 kbp DNA containing an ARS1 (or mutated) origin (chromatinized or not).** Dataset acquired as described in Fig. 2a. **a** Traces collected on DNA containing a chromatinized origin illustrating motion of ORC^JF646: (i) initially localized within the bin containing the chromatinized origin or (ii) initially localized elsewhere. **b** Histograms of the diffusion constants of ORC^JF646 on DNA molecules containing (i) a non-chromatinized origin, (ii) a chromatinized origin, (iii) a non-chromatinized mutated origin, and (iv) a chromatinized mutated origin. Only foci containing 1 or 2 ORC^JF646 are included in the analysis ($N_{foci}$). Data are presented as mean values ± one-sigma Wilson confidence intervals. The observed bimodal distribution is fitted to a double log-normal function (solid black line) which identifies a slow ORC population (55% of the distribution with 0.0049 ± 0.0028 kbp²/s, mean ± SEM) and a fast ORC population (0.152 ± 0.023 kbp²/s, mean ± SEM). The dashed line indicates the average of the means of the two normal distributions used to fit $\log_{10}(D)$; this average is used as a threshold

($D^*$). The means of these subpopulations are imposed in the fits to the data in (ii–iv). Created with BioRender.com. **c** Examination of the slow ORC population. (i) Fitted proportion of the slow ORC population for the datasets in (**b**-i)-(**b**-iv), as indicated by the corresponding color bar (ARS1: non-mutated origin; Mut ori: mutated origin; nuc: chromatinized). (ii) The population of ORC initially localized within the bin containing the (chromatinized or not) origin was extracted from the dataset; the fraction of this population that is slow ($D < D^*$) is shown here. (iii) The population of ORC initially localized outside the bin containing the (chromatinized or not) origin was extracted from the dataset; the fraction of this population that is slow ($D < D^*$) is shown here. $N_{foci-ARS1} = 69$; $N_{foci-ARS1+nuc} = 35$; $N_{foci-Mut\ ori} = 40$; $N_{foci-Mut\ ori+nuc} = 27$. Data are presented as mean values ± one-sigma Wilson confidence intervals. Statistical significance is determined by a two-sided binomial test: n. s. not significant, * $p < 0.05$, ** $p < 0.005$, **** $p < 0.00005$, ***** $p < 0.000005$ ****** $p < 0.0000005$. Source data are provided as a Source Data file.

of MCM[5,15]. When performed with ORC^JF646 and Cdc6 only (see next section for experiments involving MCM), these experiments provided a read-out of long-lived, stable ORC binding. Concretely, we incubated ORC^JF646 and Cdc6 in bulk, for 30 min with the four types of DNA molecules (untethered to beads, hence with free ends), prior to imaging in single-molecule conditions. We separately verified that an extended incubation period alone did not affect the distributions of H2A^AF488 position or stoichiometry on chromatinized DNA (Supplementary Fig. 4.1). We then introduced these pre-incubated DNA molecules into the flow cell, trapped, them, and imaged stable ORC^JF646 molecules that remained bound to the DNA (Fig. 4a).

Following this experimental approach, on bare DNA molecules containing the ARS1 origin we observed either DNA molecules devoid of ORC^JF646 ($N_{DNA-no\ ORC} = 10$, 21% of total), or DNA molecules ($N_{DNA-ORC} = 38$, 79% of total) with ORC^JF646. Using the latter, we plotted the spatial distribution of ORC^JF646 (Fig. 4b-i). We found that 61% of the ORC^JF646 foci were localized in the origin bin ($N_{foci} = 38$, $N_{foci\ in\ origin} = 23$), which emphasizes the preference of stable ORC^JF646 binding at the origin compared to the other DNA sequences in the 10.4 kbp DNA[15]. Stoichiometric analysis showed no difference with the experiments reporting on rapid ORC^JF646 binding (compare Supplementary Fig. 2b-i, Supplementary Fig. 4.2b-i). On DNA molecules with a chromatinized ARS1 origin, the overall fraction of DNA molecules

containing ORC^JF646 foci was reduced by a factor of two, to 41% ($N_{DNA-no\ ORC} = 74$, $N_{DNA-ORC} = 52$). However, we observed a similar fraction (51%) of ORC^JF646 foci localized in the origin bin ($N_{foci} = 43$, $N_{foci\ in\ origin} = 22$, Fig. 4b-ii), indicating that chromatinization of ARS1 did not enhance the likelihood of finding ORC^JF646 stably bound in the origin bin relative to elsewhere on the DNA. Interestingly, stoichiometric analysis showed that such conditions led to an increase in the number of ORC^JF646 molecules per focus (Supplementary Fig. 4.2b-ii). When these experiments were repeated on DNA molecules with the mutated origin, we predominantly found DNA molecules devoid of ORC^JF646, irrespective of whether nucleosomes were present ($N_{DNA-no\ ORC} = 45$, $N_{DNA-ORC} = 2$, Fig. 4b-iv) or not ($N_{DNA-no\ ORC} = 32$, $N_{DNA-ORC} = 10$, Fig. 4b-iii). This suggested that stable ORC binding probed under these experimental conditions is predominantly associated with DNA sequence. For the four conditions probed, ORC^JF646 binding in the bin containing the origin (chromatinized or not) was subjected to tests of statistical significance (Fig. 4c). These results indicate that statistically significant enhancement of stable ORC binding depends on the presence of ARS1, but not on the presence of nucleosomes.

Previous studies have shown direct interactions between ORC and nucleosomes, suggesting a potential for nucleosome remodeling by ORC[11,24]. Thus, we assessed whether, in our experiments, the presence of ORC^JF646 and Cdc6 impacted the spatial distribution and

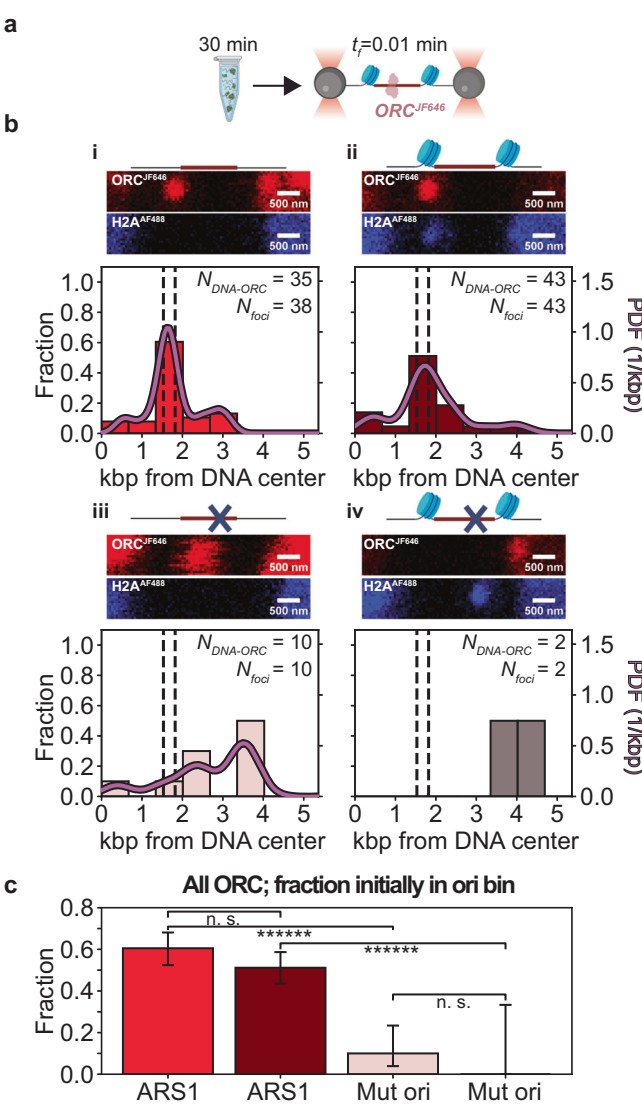

**Fig. 4 | Spatial distribution of stably bound ORC$^{JF646}$ on 10.4 kbp DNA containing an ARS1 (or mutated) origin (chromatinized or not). a** ORC$^{JF646}$ and Cdc6 are incubated with DNA molecules for 30 min with ATP. Then, the DNA-protein complex is flushed into the single-molecule flow cell, tethered, and transferred to the buffer channel for imaging; $t_f$: frame acquisition rate. Created with BioRender.com. **b** (i-iv) Schematics of the origin regions, representative fluorescence scans over the full length of the DNA (red: ORC$^{JF646}$; blue, H2A$^{AF488}$; note that DNA molecules can be captured in opposite orientations), and spatial distributions of ORC$^{JF646}$ immediately after introduction into the flow cell. (i) ORC$^{JF646}$ spatial distribution on non-chromatinized DNA, deduced from red diffraction-limited spots ($N_{foci}$) using 35 DNA molecules ($N_{DNA-ORC}$). The dashed lines indicate the location of the NPSs, and the solid curve indicates the kernel density estimation of the data (PDF: probability density function). (ii) ORC$^{JF646}$ spatial distribution on chromatinized DNA, deduced from red diffraction-limited spots ($N_{foci}$) using 43 DNA molecules ($N_{DNA-ORC}$). (iii) ORC$^{JF646}$ spatial distribution on non-chromatinized mutated DNA, deduced from red diffraction-limited spots ($N_{foci}$) using 10 DNA molecules ($N_{DNA-ORC}$). (iv) ORC$^{JF646}$ spatial distribution on chromatinized mutated DNA, deduced from red diffraction-limited spots ($N_{foci}$) using 2 DNA molecules ($N_{DNA-ORC}$, note that the total number of DNA molecules analyzed with and without ORC$^{JF646}$ is 47, see Supplementary Fig. 4.3). The dashed lines indicate the location of the NPSs, and the solid curve indicates the kernel density estimation of the data (PDF: probability density function). Created with BioRender.com (**c**) ORC$^{JF646}$ occupancy probability for the bins containing the origin in (**b**-i)–(**b**-iv), as indicated by the corresponding color bar (ARS1: non-mutated origin; Mut ori: mutated origin; nuc: chromatinized). $N_{foci-ARS1}$ = 38; $N_{foci-ARS1+nuc}$ = 43; $N_{foci-Mut\,ori}$ = 10; $N_{foci-Mut\,ori+nuc}$ = 2. Data are presented as mean values ± one-sigma Wilson confidence intervals. Statistical significance is determined by a two-sided binomial test ($p$ value$_{ARS1/ARS1+nuc}$ = 2.6 × 10$^{-1}$; $p$ value$_{ARS1/Mut\,ori}$ = 3.4 × 10$^{-14}$; $p$ value$_{ARS1+nuc/Mut\,ori+nuc}$ = 0.0; $p$ value$_{Mut\,ori/Mut\,ori+nuc}$ = 1.0). n.s. not significant, ****** $p$ < 0.0000005. Source data are provided as a Source Data file.

## Nucleosomes flanking the origin permit MCM recruitment but restrict subsequent motion

Having probed both rapid and stable ORC binding and established that the presence of ARS1 was beneficial to both but the presence of nucleosomes only to the former, we wanted to test how the presence of nucleosomes impacted the loading of MCM. We, therefore, set out to probe the spatial positioning, stoichiometry, and dynamics of MCM on a chromatinized origin. To visualize MCM, we used JF646-labeled MCM in loading reactions. MCM was labeled by introducing a HaloTag on the N-terminus of its Mcm3 subunit, hereafter referred to as MCM$^{JF646}$, and MCM$^{JF646}$ performed normally in a bulk loading assay (Supplementary Fig. 1.1). We then incubated ORC, Cdc6 and MCM$^{JF646}$/Cdt1 with DNA in bulk, without tethering to beads, for 30 min[5,15] as described above for the experiments that probed for stable ORC$^{JF646}$ binding. We next introduced these pre-incubated DNA molecules into the flow cell, and imaged MCM$^{JF646}$ as described above (Fig. 5a).

In the absence of nucleosomes, MCM$^{JF646}$ foci were broadly distributed about the ARS1 origin (Fig. 5b-i), as previously described[15]. Chromatinization of the ARS1 origin (formed with H2A$^{AF488}$, as above), however, resulted in a strikingly different situation: MCM$^{JF646}$ foci were now mainly localized in the origin bin (Fig. 5b-ii). Repeating this experiment on bare DNA containing the mutated origin again resulted in a broad distribution of MCM$^{JF646}$ (Fig. 5b-iii), this time with a slight peak in the same bin as was observed in the experiments that probed rapid ORC$^{JF646}$ binding (Fig. 2b-iii). Chromatinization of this mutated origin did not significantly increase the presence of MCM$^{JF646}$ in the origin bin (Fig. 5b-iv). For the four types of DNA molecules probed, MCM$^{JF646}$ binding in the bin containing the origin (chromatinized or not) was subjected to tests of statistical significance (Fig. 5c). These results indicate that chromatinization of the ARS1 origin, but not of a DNA segment without ARS1, statistically significantly increases the population of MCM bound there.

We also examined the stoichiometry of MCM$^{JF646}$ foci bound to four different types of DNA molecules tested under these four conditions. Stoichiometric analysis showed that the MCM$^{JF646}$ foci

stoichiometry of H2A$^{AF488}$. For reference, following a 30 min buffer-only bulk incubation of the DNA with a chromatinized ARS1 origin ($N_{DNA}$ = 51), 80% of H2A$^{AF488}$ foci were found in the origin bin ($N_{foci}$ = 45, $N_{foci\,in\,origin\,bin}$ = 36), and the best fit of the H2A$^{AF488}$ stoichiometry distribution yielded $p_{h2a}$ = 0.73 and $p_{occupancy}$ = 0.98 (Supplementary Fig. 4.1), values similar to those obtained without such bulk incubation (Fig. 1c). When the bulk incubation was performed in the presence of ORC$^{JF646}$ and Cdc6 ($N_{DNA}$ = 126), however, the fraction of H2A$^{AF488}$ foci in the origin bin was reduced to 49% ($N_{foci}$ = 134, $N_{foci\,in\,origin\,bin}$ = 66) (Supplementary Fig. 4.3b), and the best fit of the H2A$^{AF488}$ stoichiometry distribution yielded $p_{h2a}$ = 0.81 and $p_{occupancy}$ = 0.52. The latter parameters suggested a maintenance of H2A$^{AF488}$-H2B dimer stability but a less complete NPS occupancy, likely due to nucleosome displacement, as evidenced by the presence of nucleosomes in other DNA regions. When the bulk incubation in the presence of ORC$^{JF646}$ and Cdc6 was performed on DNA molecules with a chromatinized mutated origin ($N_{DNA}$ = 47), an intermediate value of 64% of H2A$^{AF488}$ foci were found in the origin bin ($N_{foci}$ = 56, $N_{foci\,in\,origin\,bin}$ = 36) (Supplementary Fig. 4.3c), and the best fit of the H2A$^{AF488}$ stoichiometry distributions of H2A$^{AF488}$ yielded $p_{h2a}$ = 0.60 and $p_{occupancy}$ = 0.72. The latter parameters indicated a reduction in H2A$^{AF488}$-H2B dimer stability without a change in NPS site occupancy. In summary, these results show that under these more extended incubation conditions, the presence of ORC and Cdc6 can influence nucleosome positioning and stability.

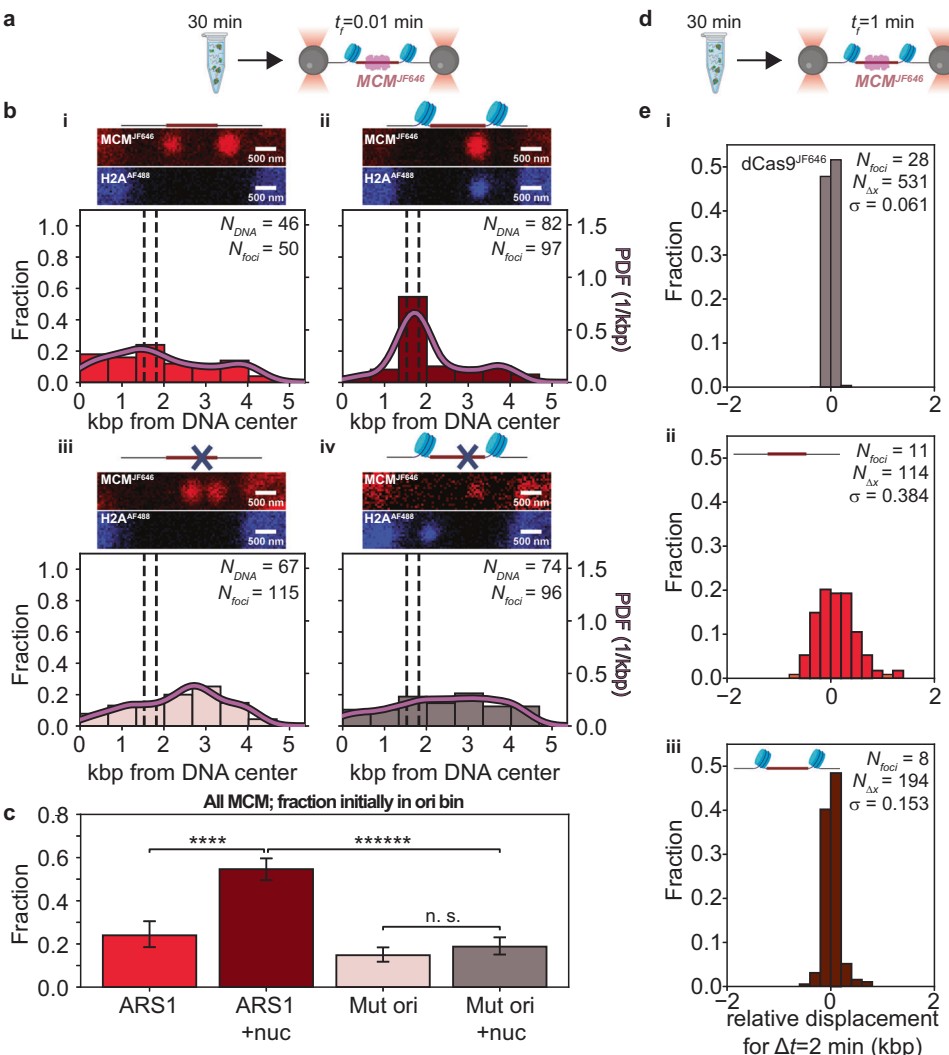

**Fig. 5 | Nucleosomes spatially constrain MCM$^{JF646}$ at the origin. a** ORC, Cdc6 and MCMJF646/Cdt-1 are incubated with DNA molecules for 30 min with ATP. Then, the DNA-protein complex is flushed into the single-molecule flow cell, tethered, and transferred to the buffer channel for imaging; $t_f$: frame acquisition rate.
**b** Schematics of the origin regions, representative fluorescence scans over the full length of the DNA (red: MCM$^{JF646}$; blue, H2A$^{AF488}$; note that DNA molecules can be captured in opposite orientations), and spatial distributions of MCM$^{JF646}$ acquired immediately after introduction into the flow cell. (i) MCM$^{JF646}$ spatial distribution on non-chromatinized DNA. The dashed lines indicate the location of the NPSs, and the solid curve indicates the kernel density estimation of the data (PDF: probability density function). (ii) MCM$^{JF646}$ spatial distribution on chromatinized DNA. (iii) MCM$^{JF646}$ spatial distribution on non-chromatinized mutated DNA. (iv) MCM$^{JF646}$ spatial distribution on chromatinized mutated DNA. **c** MCM$^{JF646}$ occupancy probability for the bins containing the (chromatinized or not) origin for the datasets in

(**b**-i)−(**b**-iv), as indicated by the corresponding color bar (ARS1: non-mutated origin; Mut ori: mutated origin; nuc: chromatinized). $N_{foci-ARS1} = 50$; $N_{foci-ARS1+nuc} = 97$; $N_{foci-Mut ori} = 115$; $N_{foci-Mut ori+nuc} = 96$. Data are presented as mean values ± one-sigma Wilson confidence intervals. Statistical significance is determined by a two-sided binomial test ($p$ value$_{ARS1/ARS1+nuc} = 1.3 \times 10^{-5}$; $p$ value$_{ARS1/Mut ori} = 7.3 \times 10^{-2}$; $p$ value$_{ARS1+nuc/Mut ori+nuc} = 1.7 \times 10^{-14}$; $p$ value$_{Mut ori/Mut ori+nuc} = 3.4 \times 10^{-1}$): n.s. not significant, **** $p < 0.00005$ and ****** $p < 0.0000005$. **d** ORC, Cdc6, and MCM$^{JF646}$/Cdt-1 are incubated with DNA molecules as in (**a**) but at an acquisition frequency of one frame per minute ($t_f = 1$ min). **e** Histograms of the relative displacements between 2 min intervals ($\Delta t = 2$ min) for (i) dCas9$^{JF646}$, (ii) for MCM$^{JF646}$ on DNA containing a non-chromatinized origin or (iii) chromatinized origin. $N_{foci}$, number of molecules; $N_{\Delta x}$, number of relative displacements; $\sigma$, standard deviation. Source data are provided as a Source Data file. Created with BioRender.com.

predominantly contained 1 or 2 molecules for all cases (Supplementary Fig. 5.1). However, foci found on DNA molecules that contained the chromatinized ARS1 origin exhibited the highest fraction of foci in the origin bin that contained two MCM$^{JF646}$ molecules (53%, right plot in Supplementary Fig. 5.1-ii). We discuss this further below, together with the overall observation that the spatial distributions of MCM resembled those obtained for rapid ORC$^{JF646}$ binding (Fig. 2) more closely than those obtained for stable ORC$^{JF646}$ binding (Fig. 4).

As our experiments showed that the presence of ORC$^{JF646}$ and Cdc6 during 30 min incubations in bulk impacted nucleosome positioning and stability (Supplementary Fig. 4.3), we also monitored nucleosome positioning and stability upon the inclusion of

MCM$^{JF646}$. On DNA molecules with a chromatinized ARS1 ($N_{DNA} = 110$), 83% of H2A$^{AF488}$ foci were found in the origin bin ($N_{foci} = 81$, $N_{foci in origin bin} = 67$), and the best fit of the H2A$^{AF488}$ stoichiometry distribution yielded $p_{h2a} = 0.84$ and $p_{occupancy} = 0.89$ (Supplementary Fig. 5.2b), which are all values similar to those obtained following incubation with buffer alone. However, on DNA with a chromatinized mutated origin ($N_{DNA} = 123$), only 50% of H2A$^{AF488}$ foci were located in the origin bin ($N_{foci} = 70$, $N_{foci in origin bin} = 140$), and the best fit of the H2A$^{AF488}$ stoichiometry distribution yielded $p_{h2a} = 0.91$ and $p_{occupancy} = 0.40$ (Supplementary Fig. 5.2c). The latter parameters suggested a maintenance of H2A$^{AF488}$-H2B dimer stability but a less complete NPS site occupancy, likely due to

nucleosome displacement, as evidenced by the presence of nucleosomes in other DNA regions. We speculate that increased MCM[JF646] loading on a chromatinized ARS1 origin (Fig. 5b-ii) relative to the chromatinized mutated origin (Fig. 5b-iv) limits the continued access of ORC and Cdc6 to nucleosomes, and hence their impact (see also Discussion).

Lastly, we tested whether a pair of nucleosomes surrounding ARS1 can prevent MCM, once loaded, from diffusing outwards away from ARS1. To test this directly, we monitored MCM[JF646] position over an extensive duration of time by repeated the preceding experiments but imaging MCM[JF646] at a 100-fold slower rate (one frame per min, Fig. 5d; see sample kymographs in Supplementary Fig. 5.3). This allowed us to probe for MCM[JF646] motion occurring on timescales up to ~15 min. We measured the relative displacements of MCM[JF646] foci every 2 min (or two frames) until JF646 was no longer visible due to photobleaching (lifetime of the JF646 dye on DNA-bound dCas9[JF646] under identical imaging conditions was $1.23 \times 10^3$ s). These data were benchmarked relative to the displacement distribution of dCas9[JF646] (Fig. 5e-i) obtained under identical experimental conditions, which was centered about 0 and had a width of $\sigma = 0.061$ kbp that derived from experimental noise. The distribution of displacements by MCM[JF646] on DNA molecules with a bare ARS1 had a substantially increased width relative to dCas9[JF646] (Fig. 5f-ii; $\sigma = 0.398$ kbp), indicating one-dimensional diffusion of the MCM[JF646] foci as previously observed[15]. However, on DNA containing the chromatinized ARS1, the distribution of displacements by MCM[JF646] again had a narrow width (Fig. 4f-iii; $\sigma = 0.153$ kbp). This indicated that one-dimensional diffusion of MCM away from the ARS1 origin of replication was hampered by the presence of flanking nucleosomes.

## Discussion

Chromatin replication starts from origins that are flanked by nucleosomes. We have studied at the single-molecule level to understand how the presence of nucleosomes impacts the binding and mobility of ORC in the vicinity of the origin and subsequently, the recruitment and mobility of MCM.

### Nucleosomes facilitate rapid binding of ORC to the origin and decrease its mobility

Our experiments first probed the rapid binding of yeast ORC to DNA following a short incubation period in the single-molecule instrument (Fig. 2) and the subsequent motion of these molecules (Fig. 3). As our previous work on bare DNA has shown, mobile ORC molecules can contribute to origin recognition by scanning the DNA–given the measured diffusion constant of the fast ORC population (Fig. 3b-i), even during a short 5 s incubation period an individual ORC molecule can scan ~1.2 kbp on bare DNA, which contributes to its observed binding at many different sites along the DNA (Fig. 2b-i). Repeating these experiments in the presence of a chromatinized origin showed that the presence of nucleosomes provides a mild increase in local ORC binding (Fig. 2b-ii, c) and a reduction in ORC mobility (Fig. 3b-ii, c). The latter can derive from spatial confinement of ORC by nucleosomes, direct interactions between ORC and nucleosomes, or a combination thereof. A reduction in ORC mobility can furthermore account for the observed increase in local ORC binding: for example, if nucleosomes similarly reduced the influx of mobile ORC molecules into the origin region and their efflux out of it, then binding from solution and/or direct interactions between ORC and nucleosomes could account for the observed increase in local ORC binding. Certainly, our data suggest that in the yeast system, ORC molecules can readily locate a chromatinized origin via binding from solution even if their one-dimensional diffusion is blocked; whether the same holds true for higher eukaryotes that lack sequence-specific origins remains to be determined.

### Stable binding of ORC is primarily enabled by the origin of replication

We next asked whether nucleosomes affected the stable binding of ORC to DNA following bulk incubation of ORC with DNA molecules over an extended duration. Examination in the single-molecule instrument revealed that a bare ARS1 origin alone sufficed to retain stably bind ORC molecules to DNA (Fig. 4a); no further increases in the yield of stable ORC were observed with a chromatinized ARS1 origin (Fig. 4b). Interestingly, while DNA molecules that contained a mutated origin could recruit rapidly bound ORC (Fig. 2d,e), they could not stably retain ORC, irrespective of whether nucleosomes surrounded the origin (Fig. 4d,e). This most likely suggests that rapidly but non-specifically recruited ORC molecules unbind from the DNA directly into solution[25], as our direct tracking of the motion of ORC does not provide experimental evidence that ORC can bypass flanking nucleosomes (Fig. 3). These experiments thus showed that ARS1 is required for stable ORC binding[15] even in the presence of nucleosomes. This contrast with results from another study[26] that suggested that a specific sequence might not be necessary and found ORC bound specifically to nucleosomes. Such discrepancies may have resulted from differences in the experimental preparations employed. Nevertheless, our findings are in line with previous findings that show that origin sequences are required for yeast replication in vitro on chromatinized DNA and in vivo in budding yeast[27,28].

Stoichiometric analysis of either rapidly or stably bound ORC molecules bound to the region containing the ARS1 origin indicated a slight increase in ORC stoichiometry upon chromatinization (Supplementary Fig. 2). We computed the resulting ratio of 1 ORC molecule:2 ORC molecules to be ~1:3, very similar to the ratio previously determined using cryo-EM[16]. An increase in ORC stoichiometry upon chromatinization of ARS1 could result from a similar influx of ORC to the origin but a reduced efflux as the diffusion of ORC molecules away from the origin[15] is reduced by the presence of nucleosomes, or an increased influx of ORC to the origin due to direct interactions between ORC and nucleosomes[26,29], or a combination of these effects.

### Influence of ORC and MCM on nucleosome remodeling at the origin of replication

Previous studies have indicated that ORC has properties of a chromatin remodeler[12–14,24], either due to direct interaction of ORC with the nucleosomes or due to DNA bending at the origin[30]. Indeed, we confirmed that extended incubation of chromatinized substrates with ORC and Cdc6 could broaden the spatial distribution and/or reduce the stoichiometry of H2A[AF488] histones embedded within our nucleosomes (Supplementary Fig. 4.3), in agreement with published work reporting in vitro remodeling of H2A-H2B dimers by ORC[24]. Interestingly, our data monitoring the position and stoichiometry distributions of H2A[AF488] following an extended incubation with all loading factors, however, show no changes therein on DNA containing a chromatinized ARS1 origin (Supplementary Fig. 5.2). This suggests a protective role for MCM in the maintenance of epigenetic chromatin states prior to the firing of DNA replication[31].

### Loading of MCM onto a chromatinized origin

Our results showed that the lack of a replication origin inhibits the stable binding of ORC to DNA (Fig. 4), but not the rapid binding of ORC (Fig. 2) or the loading of MCM (Fig. 5). This implies that stable binding of ORC at ARS1 is not a prerequisite to MCM loading, which is consistent with biochemical experiments showing that MCM loading can occur independently of specific origins[5]. Furthermore, our measurements of the spatial distributions of MCM indicate that for the four types of DNA molecules tested, there is a probability of finding MCM molecules all along the DNA molecule. Several factors underlie this behavior. First is the one just mentioned: ORC binding is not strictly sequence-specific, even in yeast, and thus MCM loading can occur at

different DNA sequences. Indeed, the spatial distribution of MCM molecules (Fig. 5) showed similar features to the spatial distributions of rapid ORC binding (Fig. 2). Second, both bulk biochemical assays[4,5] and our previous single-molecule studies[15] have shown that loaded MCM can undergo linear diffusion on DNA. In the buffer conditions employed, the diffusion constant of MCM equals $0.0008 \pm 0.0002 \, kbp^2 \, s^{-1}$ (Fig. 5e-ii), from which we estimate that diffusion of MCM during the extended incubation period contributed ~1.4 kbp to the broadening of the observed spatial distribution.

On DNA molecules containing a chromatinized ARS1, we found that the probability of finding MCM proteins in the vicinity of the origin was increased by more than two-fold compared to that on DNA molecules containing bare ARS1. Given that this increase is larger for MCM (Fig. 5c) than for rapidly bound ORC (Fig. 2c), this data alone could suggest that the presence of nucleosomes prevents the diffusion of MCM molecules away from the origin. The lack of a similar increase in the probability of MCM foci close to the origin due to the chromatinization of a mutated origin, compare Fig. 4e to Fig. 4d (despite a corresponding mild increase in rapid ORC binding, compare Fig. 2e to Fig. 2d) is consistent with previous work showing that nucleosomes reduced non-specific loading of MCM on a mutated origin[14,32].

Stoichiometric analysis reveals that the detected MCM foci in the region of the ARS1 origin predominantly have a stoichiometry of 1 or 2. In the presence of nucleosomes, however, we clearly observed an increased fraction of foci with a stoichiometry of 2 (Supplementary Fig. 5.1b). While in our experimental configuration, a measured MCM stoichiometry of 2 within a diffraction-limited focus cannot distinguish an MCM double hexamer from two single MCM hexamers, our results are consistent with previous work showing that a chromatinized ARS1 origin favors the formation of the MCM/ORC (MO) intermediate complex and, consequently, the loading of MCM double hexamers[16].

## MCM spatially constrained to a chromatinized origin on long timescales

Our experimental tracking of MCM position over long incubation times shows that the presence of nucleosomes spatially constrains MCM (Fig. 5e). Does this spatial confinement help to explain how the presence of origin-flanking nucleosomes favors pre-RC formation[11,12]? Such spatial confinement will limit the diffusion of MCM single hexamers[15,23] and thereby reduce the likelihood of MCM double hexamer formation through the encounter of properly oriented, diffusing MCM single hexamers[23]. Possibly, this favors MCM double hexamer formation via the MO intermediate[16]. Future investigations will further investigate the confinement of MCM by nucleosomes, as it could potentially contribute to the recycling of licensing factors in cases of DNA replication stress, where reloading of factors is not possible. replisome assembly at the origins of replication flanked by nucleosomes.

Our findings provide new insights into the role of chromatin in DNA replication initiation. We demonstrate that the presence of nucleosomes surrounding origins contributes to the loading and subsequent spatial constraint of MCM hexamers there, which likely favors subsequent CMG formation and the initiation of DNA replication at replication origins. Future investigation will focus on the quantification of these downstream events. This research will additionally open up new avenues to further explore the role of chromatin in origin firing and the mechanisms of chromatin replication.

## Methods

### Biological materials: protein purification and labeling

**Histone octamers.** Plasmids pCDFduet.H2A-H2B and pET-Duet.H3-H4, expressing *S. cerevisiae* histones H2A, H2B, H3, and H4 were kindly provided by Dr. Martin Singleton (Francis Crick Institute, London). Lysine 120 of H2A was changed to cysteine by site-directed mutagenesis to facilitate fluorescent labeling. These histones were co-expressed in *Escherichia coli* strain BL21-codonplus-DE3-RIL (Agilent), and the histone octamer with the K120>C mutation in histone H2A was purified according to ref. 33. Cells were grown to a density with an $OD^{600}$ of 0.3–0.5, and expression of the histones was induced with 400 μM isopropyl 1-thio-ß-D-galactopyranoside (Santa Cruz Biotechnology Inc) for 16 h at 17 °C while shaking at 180 rpm. Cells were lysed by sonication in a Qsonica Q500 sonicator for 2 min with cycles of 5 s on and 5 s off and an amplitude of 40%, in histone lysis buffer (0.5 M NaCl, 20 mM Tris-HCl pH 8, 0.1 mM EDTA, 1 mM DTT, 0.3 mM PMSF, and protease cOmplete inhibitor). Supernatant containing the histone octamers was purified on a 5 mL Hi-Trap Heparin column (Cytiva) and eluted with a gradient of 0.5–2 M NaCl in 20 mM Tris-HCl pH 8, 0.1 mM EDTA, 1 mM DTT. Peak fractions were analyzed on a 12% SDS-PAGE, and octamer-containing fractions were further purified on a Superdex 200 increase (Cytiva) using histone GF buffer (2 M NaCl, 20 mM Tris-HCl pH 8, 0.1 mM EDTA, and 1 mM DTT). Peak fractions were analyzed on a 12% SDS-PAGE gel, and fractions containing histone octamers were pooled and concentrated in an Amicon Ultra-4 Ultracell 30 kDa centrifugal filter (Merck-Millipore #UFC803024). The protein concentration was determined with Bio-Rad Protein Assay Dye Reagent (Bio-Rad # 5000006).

**Cdc6.** *S. cerevisiae* Cdc6 protein expression was induced in BL21-CodonPlus(DE3)-RIL cells (Agilent #230245) transformed with pGEX-6P-1 wt GST-cdc6 using 400 μM IPTG for 16 h at 16 °C. Cells were harvested in Cdc6 lysis buffer (50 mM $K_XPO_4$ pH 7.6, 150 mM KOAc, 5 mM $MgCl_2$, 1% Triton X-100, 2 mM ATP, cOmplete™ EDTA-free Protease Inhibitors (Sigma-Aldrich #5056489001), 1 mM DTT, and sonicated in a Qsonica Q500 sonicator for 2 min with cycles of 5 s and 5 s off and an amplitude of 40%. After centrifugation, Cdc6 protein was purified from the supernatant by incubating for 1 h at 4 °C with glutathione beads Fastflow (GE Healthcare #17-5132-02). The beads were washed 20 times with 5 ml Cdc6 lysis buffer, and Cdc6 was released from the beads by digestion with Precision protease (GE Healthcare #27-0843-01) at 4 °C for 16 h. Subsequently, the Cdc6 eluate was diluted with Cdc6 dilution buffer (50 mM $K_XPO_4$ pH 7.6, 5 mM $MgCl_2$, 0.1% Triton X-100, 2 mM ATP, and 1 mM DTT) to a final KOAc concentration of 75 mM and incubated with hydroxyapatite Bio gel HTP (Bio-Rad #130-0402) for 45 min at 4 °C. The beads were washed five times with Cdc6 wash buffer (50 mM $K_XPO_4$ pH 7.6, 75 mM KOAc, 5 mM $MgCl_2$, 0.1% Triton X-100, 2 mM ATP, and 1 mM DTT), and then washed five times with Cdc6 rinse buffer (50 mM $K_XPO_4$ pH 7.6, 150 mM KOAc, 5 mM $MgCl_2$, 15% glycerol, 0.1% Triton X-100, and 1 mM DTT). Next, Cdc6 was eluted from the column in 1-ml fractions with Cdc6 elution buffer (50 mM KXPO4 pH 7.6, 400 mM KOAc, 5 mM $MgCl_2$, 15% glycerol, 0.1% Triton X-100, and 1 mM DTT). Finally, fractions containing Cdc6 were pooled, dialyzed twice for 1 h against Cdc6 dialysis buffer (25 mM HEPES-KOH pH 7.6, 100 mM KOAc, 10 mM MgOAc, 10% glycerol, and 0.02% NP40 substitute) in a 10 kDa cut off Slide-A-Lyzer Cassette (Thermo Scientific #66380), and concentrated in an Amicon Ultra-4 Ultracell 30 kDa centrifugal filter (Merck-Millipore #UFC803024). Aliquots were snap-frozen and stored at −80 °C. The protein concentration was determined with Bio-Rad Protein Assay Dye Reagent (Bio-rad # 5000006).

**ORC and Halo-tagged ORC.** ORC complex with a CBP-TEV tag on orc1 was purified from *S. cerevisiae* strain ySDORC, and ORC complex with a CBP-TEV-Halo tag on orc3 was purified from strain yTL158. Cells were seeded at a density of $2*10^7$ cells per ml in YP medium (1% yeast extract and 2% peptone), supplemented with 2% raffinose and grown at 30 °C and 180 rpm until a density of $3–5*10^7$ cells/ml was reached. Then cells were arrested in G1 by adding 100 ng/ml α-mating factor (Tebu-Bio #089AS-60221-5) for 3 h followed by the addition of 2% galactose for 3 h to induce the expression of ORC. Cells were harvested by centrifugation and washed with ORC lysis buffer (25 mM HEPES-KOH pH

7.6, 0.05% NP-40 substitute, 10% glycerol, 0.1 M KCl, and 1 mM DTT). After centrifugation, cells were suspended in ORC lysis buffer supplemented with protease inhibitors (cOmplete™ EDTA-free Protease Inhibitors (Sigma-Aldrich #5056489001) and 0.3 mM PMSF) and dropped into liquid nitrogen. The frozen droplets were ground in a freezer mill (6875 SPEX) for six cycles (run time 2 min and cool time 1 min with a rate of 15 cps), and the resulting powder was suspended in ORC lysis buffer supplemented with protease inhibitors. The lysate was cleared in a Beckman-Coulter ultracentrifuge (type Optima L90K with rotor TI45) for 1 h at 235.000 g at 4 °C. The cleared lysate was supplemented with CaCl$_2$ to a final concentration of 2 mM and with KCl to a final concentration of 0.3 M and then incubated for 1 h at 4 °C with washed Sepharose 4B Calmodulin beads (GE Healthcare #17-0529-01) in a spinning rotor. The beads were washed 20 times with 5 ml ORC binding buffer (25 mM HEPES-KOH pH 7.6, 0.05% NP-40 substitute, 10% glycerol, 0.3 M KCl, 2 mM CaCl$_2$, and 1 mM DTT), and the protein complex was eluted from the beads with ORC elution buffer (25 mM HEPES-KOH pH 7.6, 0.05% NP-40 substitute, 10% glycerol, 0.3 M KCl, 2 mM EDTA, 2 mM EGTA, and 1 mM DTT). ORC-containing fractions were pooled, concentrated in an Amicon Ultra-4 Ultracell 30 kDa centrifugal filter (Merck-Millipore #UFC803024), and applied to a Superose 6 increase 10/300 GL column (GE Healthcare #29-0915-96) equilibrated in ORC GF buffer (25 mM HEPES-KOH pH 7.6, 0.05% NP-40 substitute, 10% glycerol, 0.15 M KCl, and 1 mM DTT). Peak fractions were pooled and concentrated in an Amicon Ultra-4 Ultracell 30 kDa centrifugal filter (Merck-Millipore #UFC803024). Aliquots were snap-frozen and stored at −80 °C. The protein concentration was determined with Bio-Rad Protein Assay Dye Reagent (Bio-Rad # 5000006).

**MCM/Cdt1 and Halo-tagged MCM/Cdt1.** MCM/Cdt1 complex with a CBP-TEV tag on mcm3 was purified from *S. cerevisiae* strain yAM33, and MCM/Cdt1 complex with a CBP-TEV-Halo tag on mcm3 was purified from strain yTL001. Cells were grown, and MCM/Cdt1 expression was induced as described for ORC. Cells were harvested by centrifugation and washed with MCM lysis buffer (45 mM HEPES-KOH pH 7.6, 0.02% NP-40 substitute, 10% glycerol, 100 mM KOAc, 5 mM MgOAc, and 1 mM DTT). After centrifugation, cells were suspended in MCM lysis buffer supplemented with protease inhibitors (cOmplete™ EDTA-free Protease Inhibitors (Sigma-Aldrich #5056489001) and 0.3 mM PMSF) and dropped into liquid nitrogen. The frozen droplets were ground in a freezer mill (6875 SPEX) for six cycles (run time 2 min and cool time 1 min at a rate of 15 cps), and the resulting powder was suspended in MCM lysis buffer supplemented with protease inhibitors. The lysate was cleared in a Beckman-Coulter ultracentrifuge (type Optima L90K with rotor TI45) for 1 h at 235.000 g and 4 °C. The cleared lysate was supplemented with CaCl$_2$ to a final concentration of 2 mM and then incubated for 1 h at 4 °C with washed Sepharose 4B Calmodulin beads (GE Healthcare #17-0529-01) in a spinning rotor. The beads were washed 20 times with 5 ml MCM binding buffer (45 mM HEPES-KOH pH 7.6, 0.02% NP-40 substitute, 10% glycerol, 100 mM KOAc, 5 mM MgOAc, 2 mM CaCl$_2$, and 1 mM DTT), and the protein complex was eluted from the beads with MCM elution buffer (45 mM HEPES-KOH pH 7.6, 0.02% NP-40 substitute, 10% glycerol, 100 mM KOAc, 5 mM MgOAc, 1 mM EDTA, 2 mM EGTA, and 1 mM DTT). MCM/Cdt1-containing fractions were pooled, concentrated in an Amicon Ultra-4 Ultracell 30 kDa centrifugal filter (Merck-Millipore #UFC803024), and applied to a Superose 6 increase 10/300 GL column (GE Healthcare #29-0915-96) equilibrated in MCM GF buffer (45 mM HEPES-KOH pH 7.6, 0.02% NP-40 substitute, 10% glycerol, 100 mM KOAc, 5 mM MgOAc, and 1 mM DTT). Peak fractions were pooled and concentrated in an Amicon Ultra-4 Ultracell 30 kDa centrifugal filter (Merck-Millipore #UFC803024). Aliquots were snap-frozen and stored at −80 °C. Protein concentration was determined with Bio-Rad Protein Assay Dye Reagent (Bio-Rad # 5000006).

**dCas9-Halo.** Halo-tagged dCas9 protein expression was induced in BL21-CodonPlus(DE3)-RIL cells (Agilent #230245) transformed with pET302-6His-dCas9-halo (Addgene #72269) using 400 μM IPTG for 16 h at 16 °C. Cells were harvested in dCas9 lysis buffer (50 mM Na$_x$PO$_4$ pH 7.0, 300 mM NaCl and protease inhibitors (cOmplete™ EDTA-free Protease Inhibitors (Sigma-Aldrich #5056489001) plus 0.3 mM PMSF)) and sonicated in an Qsonica Q500 sonicator for 2 min with cycles of 5 s on and 5 s off and an amplitude of 40%. After centrifugation, dCas9-Halo protein was purified from the supernatant by incubating for 2 h at 4 °C with Ni-NTA agarose (Qiagen #30210). The beads were washed 10 times with 5 ml dCas9 wash buffer I (50 mM Na$_x$PO$_4$ pH 7.0 and 300 mM NaCl) and three times with dCas9 wash buffer II (50 mM Na$_x$PO4 pH 7.0, 300 mM NaCl, and 20 mM Imidazole pH 7.6), and dCas9-Halo was eluted from the agarose beads with dCas9 elution buffer (50 mM Na$_x$PO$_4$ pH 7.0, 300 mM NaCl, and 150 mM Imidazole pH 7.6). Subsequently, dCas9-Halo eluate was dialyzed twice for 1 h against dCas9-dialysis buffer (50 mM HEPES-KOH pH 7.6, 100 mM KCl, and 1 mM DTT) in a 10 kDa cut off Slide-A-Lyzer Cassette (Thermo Scientific #66380) and applied to a Hi Trap SP HP column (GE Healthcare #17-1151-01) equilibrated with dCas9 dialysis buffer. The dCas9-Halo protein was eluted from the column with dialysis buffer with a KCl gradient ranging from 100 mM up to 1 M. The dCas9-Halo-containing fractions were pooled, concentrated in an Amicon Ultra-4 Ultracell 30 kDa centrifugal filter (Merck-Millipore #UFC803024), and applied to a Superdex 200 increase 10/300 GL column (GE Healthcare #28-9909-44) equilibrated in cas9 GF buffer (50 mM HEPES-KOH pH 7.6, 150 mM KCl, and 1 mM DTT). Peak fractions were pooled and concentrated in an Amicon Ultra-4 Ultracell 30 kDa centrifugal filter (Merck-Millipore #UFC803024). Aliquots were snap-frozen and stored at −80 °C. The protein concentration was determined with Bio-Rad Protein Assay Dye Reagent (Bio-Rad # 5000006).

**dCas9-Cys.** dCas9-Cys protein expression was induced in BL21-CodonPlus(DE3)-RIL cells (Agilent #230245) transformed with plasmid 10xHis-MBP-TEV-S. pyogenes dCas9 M1C D10A C80S H840A C574S (Addgene #60815) with 400 μM IPTG for 5 h at 20 °C. Cells were harvested in dCas9-cys lysis buffer (20 mM Tris-HCl pH 8.0, 500 mM NaCl, 1 mM DTT, and protease inhibitors (cOmpleteTM EDTA-free Protease Inhibitors (Sigma-Aldrich #5056489001) plus 0.3 mM PMSF) and sonicated in an Qsonica Q500 sonicator for 2 min with cycles of 5 s on and 5 s off and an amplitude of 40%. After centrifugation, dCas9-Cys protein was purified from the supernatant by incubating for 2 h at 4 °C with Ni-NTA agarose (Qiagen #30210). The beads were washed ten times with 5 ml dCas9-Cys wash buffer (20 mM Tris-HCl pH 8.0, 250 mM NaCl, 20 mM imidazole pH 7.6), and dCas9-Cys was eluted from the agarose beads with dCas9-Cys elution buffer (20 mM Tris-HCl pH 8.0, 250 mM NaCl, 150 mM imidazole pH 7.6). Subsequently, dCas9-Cys eluate was dialyzed twice for 1 h against dCas9-Cys-dialysis buffer (20 mM HEPES-KOH pH 7.6, 150 mM KCl, and 1 mM DTT) in a 10 kDa cut off Slide-A-Lyzer Cassette (Thermo Scientific #66380) and applied to a Hi Trap SP HP column (GE Healthcare #17-1151-01) equilibrated with dCas9-Cys dialysis buffer. The dCas9-Cys protein was eluted from the column with dCas9-Cys dialysis buffer with a KCl gradient ranging from 100 mM up to 1 M. The dCas9-Cys-containing fractions were pooled, concentrated in an Amicon Ultra-4 Ultracell 30 kDa centrifugal filter (Merck-Millipore #UFC803024), and applied to a Superdex 200 increase 10/300 GL column (GE Healthcare #28-9909-44) equilibrated in cas9-Cys GF buffer (20 mM HEPES-KOH pH 7.6, 150 mM KCl, and 1 mM DTT). Peak fractions were pooled and concentrated in an Amicon Ultra-4 Ultracell 30 kDa centrifugal filter (Merck-Millipore #UFC803024). Aliquots were snap-frozen and stored at −80 °C. The protein concentration was determined with Bio-Rad Protein Assay Dye Reagent (Bio-Rad # 5000006).

## Protein labeling

Strains: To create Halo-tagged mcm3, the StuI and XmaI restriction sites in plasmid pENTR4-HaloTag (Addgene #W876-1) were changed into a silent mutation following standard cloning techniques using primers TL-019-TL-020 and TL-023-TL-024. The sequence was verified by sequencing using primers TL-021-TL-022. Then the HaloTag fragment was amplified from the mutated pENTR4-HaloTag by PCR with primers TL-025 and TL-026, which were extended with an XmaI site. This amplified HaloTag was digested with XmaI, gel-purified, and ligated into plasmid pRS306 CBP-TEV-mcm3-gal1-10 mcm2, which was digested with SgrAI and dephosphorylated with CIP, resulting in plasmid pRS306 CBP-TEV-mhalo-mcm3-gal1-10 mcm2. Proper integration of the HaloTag was confirmed by sequencing with primers (see Supplementary Table 1) TL-001, TL-002, TL-027, and TL-028. Yeast strain yTL001, which expresses MCM with a Halo-tagged mcm3, was created by linearizing plasmid pRS306 CBP-TEV-mhalo-mcm3-gal1-10-mcm2 with StuI and transforming it into yeast strain yJF21, which expresses Mcm4-7 and Cdt1 upon induction with galactose.

To create an ORC complex with a halo-tagged orc3, the CBP-TEV sites was removed from plasmid pRS306 orc1-gal1-10-orc2 through Gibson assembly (NEB #E2611L) using primers TL-441, TL-443, and TL-447. The sequence for the coding region of orc1 and orc2 was confirmed by sequencing using primers TL-084, TL-087, TL-119, and TL-136. Yeast strain yTL151, which expresses orc1, 2, 5, and 6 from a galactose-inducible promoter, was created by linearizing plasmid pRS306 orcl-gal1-10-orc2 v2 delta CBP-TEV with StuI and transforming it into yeast strain yTL070, which contains an inducible expression plasmid for orc5 and orc6.

Plasmid pRS303 CBP-TEV-halo-orc3 gal1-10 orc4 was generated by cloning the CBP-TEV-halo sequence from plasmid pRS306-CBP-TEV-halo-Pri1-Gal1-10 Pri2 into plasmid pRS303-orc3-Gal1-10 orc4 through Gibson assembly (NEB #E2611L) using primers TL-446, TL-447, TL-472, and TL-473). The sequence of CBP-TEV-halo-orc3 and orc4 was verified by sequencing using primers TL-063, TL-064, TL-449, and TL-470. Yeast strain yTL158, which expresses ORC with a halo-tagged orc3, was created by linearizing plasmid pRS303-CBP-TEV-halo-orc3-Gal1-10 orc4 with NheI and transforming it into yeast strain yTL151, which contains inducible expression plasmids for orc1, orc2, orc5, and orc6.

Labeling reactions: Proteins with HaloTag were labeled with JF646-HaloTag ligand (Promega # GA1120) by incubating the proteins with a tenfold excess of dye on ice for 0.5–1 h in the presence of 1 mM ATP. Free dye was removed by gel filtration (Superose 6 increase 10/300), and the labeling efficiency was determined to be 75% and 80% for ORC$^{JF646}$ and MCM$^{JF646}$, respectively, after estimating the total protein concentration using BSA as a protein standard (Bio-Rad Protein Assay Dye Reagent) and the labeled protein concentrations by measuring the absorbance at 646 nm spectrophotometrically. Accordingly, we cannot exclude the possibility that ~25% and 20% of the observed single ORC or single MCM populations may have been partially labeled double ORC and double MCM hexamers.

Proteins with a single cysteine were labeled with Alexa Fluor 488 C5 Maleimide (Invitrogen # A10254) by incubating the proteins with a tenfold excess of dye on ice for 2 h at pH 7 in the absence of DTT. Free dye was removed by gel filtration (Superose 6 increase 10/300), for dCas9, or by dialysis with 2 M NaCl, 20 mM Tris-HCl pH 8, and 1 mM DTT followed by spin column chromatography (Zeba 7k MWCO) for histone octamers. Labeling efficiency was determined to be 81% for H2A$^{AF488}$, after estimating the total protein concentration using BSA as a protein standard (Bio-Rad Protein Assay Dye Reagent) and the labeled protein concentrations by measuring the absorbance at 488 nm spectrophotometrically.

## DNA substrates for single-molecule imaging

To generate a biotinylated 10.4 kbp DNA molecule containing ARS1 or mutated origin flanked by nucleosome positioning sequences, we ligated three different DNA fragments prepared by PCR. The left biotinylated arm with 6.6 kbp was amplified from plasmid pDRM1 (a kind gift from Daniel Ramírez-Montero) by PCR using primers HS_BN47 and HS_BN48 (ELLA Biotech) with Platinum SuperFi II DNA Polymerase (Thermo Scientific #12361010). The right biotinylated arm with 3.3 kbp was amplified as above from plasmid pDRM1 using primers HS_BN45 and HS_BN46 (ELLA Biotech). Biotinylated PCR products were purified by standard phenol-chloroform extraction, precipitated with ethanol, and digested overnight with BsaI (NEB # R3733S). Digested biotinylated arms were purified by spin column chromatography using MicroSpin S-400 HR (Amersham # 27514001).

To generate a DNA fragment compatible for ligation containing ARS1 or mutated origin flanked by nucleosome positioning sequences, we first prepared template plasmids containing the origins. Plasmid template TL20-042 with ARS1 origin site flanked by 601 and a 603 nucleosome positioning sites[34], was prepared by cloning into MluI-digested and Antarctic−dephosphorylated plasmid pSupercos1-lambda1,2[35] of PCR fragment amplified from gBlock gene fragment (IDT) pTL013[16] using primers TL-817 and TL-818, digested with AscI. Plasmid template TL22-072 with mutated ORC binding-site, was prepared by one-step cloning using NEBuilder HiFi reaction (NEB # E5520S) into plasmid pIA146[36] linearized with HindIII (NEB # R3104S) of the fragment containing the mutated origin amplified from gBlock gene fragment (IDT)pGC218[23] with primers TL-961 and TL-964, and the fragments containing the nucleosome positioning sites 601 and 603 independently amplified from gBlock gene fragment (IDT)pTL013[16] with primers TL-958–TL-959 and TL-962–TL-963, respectively.

PCR fragments containing the origins and compatible BsaI ends were prepared from TL20-042 and TL22-072 by PCR using primers HS_BN23NPb and HS_BN26NPb (ELLA Biotech). PCR products were purified by standard phenol-chloroform extraction, precipitated with ethanol, digested overnight with BsaI (NEB # R3733S), and gel-purified.

Nucleosome assembly was carried out using salt gradient dialysis[37]. Fluorescently labeled histone octamers were mixed with DNA in High Salt Buffer (10 mM Tris pH 7.5, 2 M NaCl, 1 mM EDTA, 1 mM DTT). Samples were dialyzed for 18 h against 400 ml of High Salt Buffer and gradually supplemented with 2 L of Low Salt Buffer (10 mM Tris pH 7.6, 250 mM NaCl, 1 mM EDTA, 1 mM DTT). A final dialysis step for 1 h was performed into Zero Salt Buffer (20 mM Tris pH 7.5, 1 mM EDTA, 1 mM DTT). Fluorescently labeled histone octamer concentrations were optimized by small-scale titration and nucleosomes checked by 5% native PAGE. To test that the origin sequence is free of nucleosomes and accessible to ORC and MCM, the chromatinized construct was digested with Pst I (NEB#R0140S) and fragment size checked by 5% native PAGE. PCRs fragments with BsaI compatible ends were ligated to the chromatinized origins with T4 ligase overnight at 16 °C. The concentration of nucleosomes is maintained above 20 μg/ml to avoid dissociation due to dilution[38] using commercial nucleosomes (Epicypher #160009). Final constructs were dialyzed overnight against 25 mM HEPES pH 7.6.

## Bulk assays and single-molecule experiments: *MCM recruitment and loading reactions in bulk to test protein activity*

Loading assays were carried out as follows: 50 nM ORC (or ORC$^{JF646}$), 50 nM Cdc6, and 100 nM MCM/Cdt1 (or MCM$^{JF646}$/Cdt1) were incubated with 300 ng DNA substrate (5.8 kbp circular bead-bound ARS1-containing pSK (+)-based plasmid[39]) coupled to magnetic beads for 30 min at 30 °C with mixing at 1250 rpm in 40 μl reaction buffer (25 mM HEPES-KOH pH 7.6, 10 mM MgOAc, 100 mM KOAc, 0.02% NP40, 5% glycerol, 1 mM DTT, and 5 mM ATP or ATPγS). Beads were then washed either with high salt wash buffer (45 mM HEPES-KOH pH 7.6, 5 mM MgOAc, 0.5 M NaCl, 0.02% NP-40, 10% glycerol, 1 mM EDTA, and 1 mM EGTA) followed by low salt wash buffer (45 mM HEPES-KOH pH 7.6, 5 mM MgOAc, 0.3 M KOAc, 0.02% NP-40, 10% glycerol, 1 mM EDTA, and 1 mM EGTA), or only treated with low salt wash buffer.

Finally, beads were resuspended in 10 μl elution buffer (45 mM HEPES-KOH pH 7.6, 5 mM MgOAc, 0.3 M KOAc, 10% glycerol, and 2 mM CaCl$_2$), and DNA-bound proteins were released by MNase treatment (2 min at 30° with 700 units of MNase NEB # M0247S) and analyzed by gel electrophoresis[40].

## Single-molecule instrumentation and visualization
A hybrid instrument combining optical tweezers and confocal microscopy was used to visualize the binding of DNA and protein at the single-molecule level (Q-Trap, LUMICKS) as described[7,15] with the following variations. The instrument makes use of a customer-designed microfluidic flow cell with three inlets for injection of reaction buffers from the left and up to six inlets that are introduced orthogonally and can be used as protein reservoirs or buffer exchange locations in a temperature-controlled environment. Syringes and tubing connected to the flow cell were passivated, together with the flow cell itself, with 1 mg/ml BSA followed by 0.5% Pluronic F-127 (Sigma), each incubated for at least 30 min. Next, 20 pM of the biotinylated DNA, containing either a functional origin of replication or a mutated origin, chromatinized or not, was injected into one of the three laminar-flow-separated channels. Individual DNA molecules were trapped between two 1.76 μm diameter streptavidin-coated polystyrene beads (Spherotech) initially injected into a separate channel.

In all measurements, the stiffness of both optical traps was set to 0.3 pN/nm[41,42] The tethering of individual DNA molecules was verified by analysis of the force-extension curve obtained for each DNA molecule[43] that was used for protein visualization. During fluorescence measurements, the DNA was held at a constant tension of 2 pN and the flow was turned off, unless otherwise specified. The AF488 and JF646 dyes were illuminated with two laser lines at 488 nm (2 μW) and 638 nm (7 μW), respectively, and the fluorescence from the dyes was detected on a single photon counting detector. Two-dimensional confocal scans were performed over an area of 90 × 18 pixels, which encompasses the DNA held at a force of 2 pN and the edges of both beads. The pixel size was set to 50 × 50 nm$^2$, and the illumination time per pixel was set to 0.1 ms.

## Protein concentrations and buffers in single-molecule experiments
Incubation and visualization of DNA-protein interactions in the flow cell were performed at 30 °C. ORC$^{JF646}$ binding was conducted in reaction buffer (RB) containing 25 mM HEPES-KOH pH 7.6, 100 mM potassium glutamate, 10 mM magnesium acetate, 100 μg/mL BSA, 1 mM DTT, 0.01% NP-40-S, 10% glycerol, 5 mM ATP, with 10 nM Cdc6 and 5 nM JF646-ORC. To reduce the rate of photobleaching, we add 2 mM 1,3,5,7 cyclooctatetraene, 2 mM 4-nitrobenzylalchohol, and 2 mM TROLOX.

Preparation of DNA-protein complexes in bulk for subsequent visualization in the flow cell was done as follows: 5 nM ORC was incubated with 1 nM 10.4 kbp biotinylated DNA (chromatinized or not) at 30 °C while mixing at 800 rpm in RB with 5 mM ATP. After 5 min, 10 nM Cdc6 was added to the reaction and incubated for a further 5 min. Then, 80 nM MCM$^{JF646}$/Cdt1 was added, bringing the total reaction volume to 50 μl. Following a 30 min incubation, samples were diluted 15× in RB and injected into the microfluidic chip. ORC$^{JF646}$-Cdc6 DNA complexes were assembled in the same conditions while omitting MCM$^{JF646}$/Cdt1.

## Data analysis: particle localization in 2D scans
We use the scikit-image (v0.16.2) implementation of a Laplacian of Gaussian (LoG) spot detector. The detection radius $r_{LoG}$ is set to 6.5 pixels (312 nm) for the red channel and 5 pixels (240 nm) for the blue channel; the LoG sigma parameter is given by $\sigma_{LoG} = r_{LoG} / \sqrt{2}$. We set the detection threshold to 0.2 ADU/pixel for the red channel and 0.3 ADU/pixel for the blue channel. For subpixel localization, detected spots are projected onto the x- and y-axes and then fitted with Gaussian profiles.

## Particle tracking
The spots are tracked through subsequent frames using our own implementation of the Linear Assignment Problem method[44]. We used a maximum spot linking distance of 10 pixels (480 nm) for the red channel and 4 pixels (192 nm) for the blue channel using a maximum frame gap of three frames (1.8 s) without splitting. The LAP is solved with the scipy (v1.6.1) linear sum assignment optimizer[45]. Two spots are considered colocalized if they are on average less than 4 pixels (200 nm) apart over the first five frames; spot intensities are calculated by taking the total ADU count within the detection radius.

## Location calibration
Using a reference dCas9 dataset, we map the location of two known DNA sequences onto pixel coordinates, relative to the left bead (whose coordinate is given in microns by the C-Trap metadata, as measured in the brightfield view). With these pixel coordinates, we calculate the inverse transformation, from pixel coordinate to DNA sequence location in base pairs, giving us a brightfield-to-confocal offset value (4.5 pixels) and a pixel size value (48 nm per pixel).

## Fluorophore intensity calibration
The reference dCas9 dataset is also used for fluorophore intensity calibration. Foci with one bleaching step are used to make a distribution of intensity bleaching steps and calculate the mean $\mu_{\Delta I}$ and standard deviation $\sigma_{\Delta I}$ of that distribution. The minimum bleaching step size, which is needed for stoichiometry determination of further experiments, is set to $\Delta I_{min} \leq \mu_{\Delta I} - 2\,\sigma_{\Delta I}$ in order to capture at least 95% of all bleaching events. The measured values of $\mu_{\Delta I}$ and $\sigma_{\Delta I}$ for each color are given in Table 1.

## Stoichiometry determination
To determine the number of fluorophores present within each detected spot, we perform photobleaching step counting using Change-Point Analysis (CPA). We use the ruptures (v1.1.6)[46] implementation with an L2 cost function to detect mean shifts in the signal. The minimum segment length is set to 2 and the penalty is set to $\Delta I_{min}^2$. If any steps larger than $\Delta I_{min}$ are present after the fit, the smallest steps are combined together ("pruned") until only steps larger than $\Delta I_{min}$ are left.

## Data filtering
The resulting data table of traces with number of fluorescent proteins per spot was filtered in order to reduce noise, outliers, and data that is not suitable for further motion analysis:

Diffraction-limited spots containing more than five fluorescent proteins, likely aggregates, are filtered out.

Any traces starting or ending within 1 kbp from a bead are filtered out to prevent any proteins likely stuck to a bead from entering the dataset.

Any traces starting after frame 5 are also filtered away because we do not expect any fluorescent protein to land on the DNA during the scan.

Finally, only traces with a length of five frames or more are retained for diffusion analysis.

**Table 1 | Fluorophore properties, calibrated using dCas9 data**

| Fluorophore | Signal | $\mu_{\Delta I}$ (ADU) | $\sigma_{\Delta I}$ (ADU) | $\Delta I_{min}$ (ADU) | $N_{foci}$ |
|---|---|---|---|---|---|
| AF488 | blue | 41.6 | 9.8 | 20 | 43 |
| JF646 | red | 176.7 | 22 | 75 | 14 |

## Spatial distribution analysis

Spatial distribution plots show the average position of a fluorescent spot over the first three frames. Under our imaging conditions, the localization error for a single fluorophore is ~100 bp (as determined with dCas9$^{AF488}$ and dCas9$^{JF646}$). There is also an absolute uncertainty in determining the exact position of a fluorophore on the DNA, which is related to the uncertainty in the localization of the DNA end (which can also be determined with dCas9$^{AF488}$ and dCas9$^{JF646}$). This is taken into account in our spatial distribution plots whose $x$-axes are genomic location. The bin size of the histogram is conservatively set to 670 bp to encompass the (chromatinized) origin into a single bin and is close to (but slightly larger than) the diffraction limit. Together with the binned data, we plot the kernel density estimation of the data with a bandwidth that is equivalent to half of the bin size in the histogram (335 bp) and is higher than the localization error in the imaging conditions (100 bp).

## Motion analysis

We analyzed the mean squared displacement of individual tracked foci as a function of the delay time between frames as previously described[15,47]. We employ a Gaussian Mixture Model to fit the distribution of log($D$) in the data set without a chromatinized origin, in order to differentiate between different kinetic populations. We assess the statistical preference for either a two-state or single-state model using the Bayesian Information Criterion, and the two-state model is favored. We identified these states as a static population ($D_{slow} = 0.0049 \pm 0.0028$ kbp$^2$ s$^{-1}$ (mean $\pm$ SEM)) and a diffusive population ($D_{fast} = 0.152 \pm 0.023$ kbp$^2$ s$^{-1}$). The means of these subpopulations are imposed in the fits to the other datasets.

We calculated the mean, variance, and standard error ($m$, $var$, $SEM$) of the $i$th diffusion coefficient distribution from the fitted log-normal parameters $\mu_i$ and $\sigma_i$ according to:

$$m_i = e^{\ln(10)\mu_i + (\ln(10)\sigma_i)^2}/2 \tag{1}$$

$$var_i = e^{2\ln(10)\mu_i + (\ln(10)\sigma_i)^2/2}\left(e^{(\ln(10)\sigma_i)^2} - 1\right) \tag{2}$$

$$SEM_i = \sqrt{var_i/N_i}, \tag{3}$$

where the factors of log($e$) and ln(10) account for our use of base 10, and $N_i$ is the number of values in the data set times the area of the $i$th fitted peak. It is worth noting that the dependence of $m_i$ on both $\mu_i$ and $\sigma_i$ yields mean diffusion constants larger than might be inferred by simply computing $10^{\mu_i}$.

## Probability model for H2A stoichiometry distributions

We formulate a probability model for H2A stoichiometry based on three probabilistic quantities, the NPS site occupation probability $p_{occ}$, the H2A binding probability $p_{H2A}$, and the labeling efficiency $p_{lab}$.

The probability of having 0, 1, or 2 occupied sites ($N_{occ}$) on the DNA depends on the NPS site occupation probability ($p_{occ}$) following the binomial distribution:

$$\begin{aligned} p(N_{occ} = 0) &= (1 - p_{occ})^2 \\ p(N_{occ} = 1) &= 2p_{occ}(1 - p_{occ}) \\ p(N_{occ} = 2) &= p_{occ}^2 \end{aligned} \tag{4}$$

An H2A-H2B dimer binds to a tetrasome with a probability of $p_{H2A}$. Again following the binomial distribution, the probabilities of an occupied site containing a tetrasome, hexasome, and or full nucleosome are, therefore:

$$\begin{aligned} p_{tet} &= (1 - p_{H2A})^2 \\ p_{hex} &= 2p_{H2A}(1 - p_{H2A}) \\ p_{nuc} &= p_{H2A}^2 \end{aligned} \tag{5}$$

Combining the two parts listed above yields the probabilities for encountering the possible numbers of H2A ($N_{H2A}$) on the DNA:

$$\begin{aligned} p(N_{H2A} = 1) &= p(N_{occ} = 1) \times p_{hex} + p(N_{occ} = 2) \times 2p_{tet}p_{hex} \\ p(N_{H2A} = 2) &= p(N_{occ} = 1) \times p_{nuc} + p(N_{occ} = 2) \times (p_{hex}^2 + 2p_{tet}p_{nuc}) \\ p(N_{H2A} = 3) &= p(N_{occ} = 2) \times 2p_{hex}p_{nuc} \\ p(N_{H2A} = 4) &= p(N_{occ} = 2) \times p_{nuc}^2 \end{aligned} \tag{6}$$

Finally, we need to take labeling efficiency $p_{lab}$ into account. This gives us the probabilities for finding $N_{vis}$ visible fluorophores in a diffraction-limited spot on the DNA:

$$\begin{aligned} p(N_{vis} = 1) &= p(N_{H2A} = 1) \times p_{lab} + p(N_{H2A} = 2) \times 2p_{lab}(1 - p_{lab}) \\ &\quad + p(N_{H2A} = 3) \times 3p_{lab}(1 - p_{lab})^2 \\ &\quad + p(N_{H2A} = 4) \times 4p_{lab}(1 - p_{lab})^3 \\ p(N_{vis} = 2) &= p(N_{H2A} = 2) \times p_{lab}^2 + p(N_{H2A} = 3) \times 3p_{lab}^2(1 - p_{lab}) \\ &\quad + p(N_{H2A} = 4) \times 6p_{lab}^2(1 - p_{lab})^2 \\ p(N_{vis} = 3) &= p(N_{H2A} = 3) \times p_{lab}^3 + p(N_{H2A} = 4) \times 4p_{lab}^3(1 - p_{lab}) \\ p(N_{vis} = 4) &= p(N_{H2A} = 4) \times p_{lab}^4 \end{aligned} \tag{7}$$

## Force analysis

To quantify unwrapping of DNA from nucleosomes, we performed a transformation of force-distance curves to contour length space. The persistence length $L_p$ and stretch modulus $S$ of dsDNA in the measurement buffer were determined by pulling on bare dsDNA molecules to the overstretching regime at a constant pulling speed of 100 nm/s and fitting the force-distance curves using the extensible worm-like chain model[48]:

$$d = L_c\left[1 - \frac{1}{2}\left(\frac{k_B T}{FL_p}\right)^{\frac{1}{2}} + \frac{F}{S}\right], \tag{8}$$

where $d$ is the distance between the two ends of the dsDNA, $L_c$ is the DNA contour length, $k_B$ is Boltzmann constant, and $F$ is the pulling force. The fitting results are $L_p = 44.2$ nm and $S = 1421$ pN.

The force-distance curves obtained by pulling on the chromatin sample were transformed to contour length space by calculating the DNA contour length $L_c$ using Eq. 9, an inversion of Eq. 8, and plotting $L_c$ against pulling force applied to the chromatin:

$$L_c = \frac{1}{d}\left[1 - \frac{1}{2}\left(\frac{k_B T}{FL_p}\right)^{\frac{1}{2}} + \frac{F}{S}\right]. \tag{9}$$

This results in graphs with segments of constant $L_c$. CPA (again using the ruptures library with an L2 cost function) was applied to the $L_c$-$F$ plots to identify the plateaus and extract contour length increments between the plateaus, which resulted from unwrapping of nucleosomes, as well as the nucleosome unwrapping forces. This process follows the same steps as the fluorophore bleaching step fitting analysis, but here we set the minimum step size to 10 nm, after calibration with dsDNA.

## Experiment automation

To automate some of the C-trap measurements, we used the Lumicks Harbor experiment automation scripts (https://harbor.lumicks.com/scripts) as a starting point; most notably, Joep Vanlier's automation script for catching beads, fishing for DNA, and making force-distance curves. We added the functionality to automatically acquire confocal images after successfully trapping DNA.

## Error determination

To compute the statistical error in population, stoichiometry and diffusion plots, we use the Wilson confidence interval. This is an improvement over the normal approximation for the error of sample proportion, especially for a small number of trials[49]. We use the statsmodels (Python) implementation for its calculation[50].

## Figure schematics

Figure schematics were generated using BioRender.com (Standard Academic License).

## Reporting summary

Further information on research design is available in the Nature Portfolio Reporting Summary linked to this article.

## Data availability

Source data are provided with this paper. Raw and processed ensemble and single-molecule data generated in this study have been deposited in the 4TU data repository and can be found at https://doi.org/10.4121/c5f64852-b82d-41e5-9efc-1296e6698009. Source data are provided with this paper.

## Code availability

All the code used in the current study is available at https://gitlab.tudelft.nl/nynke-dekker-lab/public/chromatin-loading-notebooks.

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

## Acknowledgements

We thank Anne Early and Lucy Drury for providing yeast strains, Nerea Murugarren for assistance in loading bulk biochemical assays, Fiona Horne and Emilie van Vet for preliminary chromatin assembly experiments, and Alessandro Costa for useful discussions. Z.L. acknowledges the support given by EMBO Postdoctoral Fellowship (ALTF 484-2022). N.D. acknowledges funding from the Netherlands Organization for Scientific Research (NWO) through Top grant 714.017.002), "BaSyC—Building a Synthetic Cell" Gravitation grant (024.003.019) of the Netherlands Ministry of Education, Culture and Science (OCW), and from the European Research Council through an Advanced Grant (REPLICHROMA; grant number 789267).

## Author contributions

H.S., J.D., and N.D. conceived the study, and N.D. supervised the study. HS designed the experiments with input from Z.L., E.v.V., J.D., and N.D. J.D. provided cell strains and advised on protein purification and biochemical conditions. T.v.L. engineered DNA substrates, purified the proteins, and performed bulk biochemical assays. H.S. labeled the proteins and assembled chromatin substrates. H.S. performed all single-molecule experiments. Z.L and E.v.V. designed and wrote data analysis routines. H.S. and E.v.V. also performed data analysis, with input from Z.L. and N.D. All authors were involved in the discussion of the data. H.S. and N.D. wrote the manuscript with input from the other authors.

## Competing interests

The authors declare no competing interests.
