## [Peer Review File · Nature Communications]

A chromatinized origin reduces the mobility of ORC and MCM through interactions and spatial constraintREVIEWER COMMENTS

Reviewer #1 (Remarks to the Author):

The manuscript of Sánchez et al., uses an in vitro single-molecule approach to study a chromatinized origin of replication. They find that ORC binds the origin with similar efficiency, whether chromatinized or not, but its mobility is reduced by nucleosomes. MCM recruitment on a chromatinized origin is shown to be efficient, but its subsequent movement away from the origin is limited. Together these results suggest that chromatinized origins in yeast are crucial for local MCM retention and thus might facilitate replisome assembly.

The study is very well executed and clear. The consistent experiments with the four different DNA constructs with/without the specific binding site and with/without the nucleosomes present give a very clear data set that in turn lead to significant conclusions. They leverage the single molecule instrumentation in a smart way and provide nice statistics with their data to underpin their conclusions. All in all, it makes me very enthusiastic about their results and I think that the manuscript is very suitable for Nature communication.

I have only a few minor comments that should be addressed:

- The binning choice for the localization of the proteins on the DNA is not clear to me. They choose it to be the diffraction limit but there is no need to link it to that because the localization precision can be better than the diffraction limit.
- So that leads to the question what really limits the size of the bin in these experiments?
- And is it possible to reduce the bin size in order to get more spatial resolution?

Reviewer #2 (Remarks to the Author):

Review of "A chromatinized origin reduces the mobility of ORC and MCM through interactions and spatial constraint"

This work from the Dekker lab investigates the role of nucleosomes flanking origin sequences on ORC and MCM binding as well as their interactions with the nucleosomes using a single molecule technique. Overall, the data does back up the claims that nucleosomes can facilitate ORC binding at the origin and decrease ORC mobility. The data also shows that the nucleosomes spatially constrain MCMs that may be loaded at the origin. It was also nice that the authors controlled for the effect of the origin sequence or nucleosomes in combination. While the text is well written, the figures should be improved upon. Figures were often not very clear, could not stand alone, and/or sometimes did not directly show the main point referenced in the text. Another major issue was the lack of any representative kymographs for any of the experimental conditions except for the development of the assay. It was also unclear if the exact assay, using labelled histones described in the first part of the results, was consistently used throughout the paper and why the authors potentially chose different experimental set ups. The discussions on the implications of chromatinized DNA favouring CMG formation and the initiation of DNA replication at sequence specific origins is a promising hypothesis.

Revision points as they came up in text or figures:

1. Minor: Use of the word 'attendant' as an adjective in abstract (line 22) and in the introduction (line 40) is confusing. Whilst it is used correctly, it is not a commonly used adjective or often used in adjective form and can be more easily understood with synonyms like resulting or consequent. In line

22 it could even be omitted.

2. Minor: What were the known standards for determining labelling efficiency for histones to determine degree of labelling to be $\text{label} = 0.81$ (Line 83)? In Line 84 the reference to methods and statement: as determined from bulk experiments is not clear. Or maybe more details in methods on how to determine labelling efficiency – how do they determine the protein concentrations and fluorophore concentrations against said standards?

3. Medium: Figure 1c left panel makes no explanation of the right y-axis. What is this “PDF (1/kbp)”? I assume it some probability density function, but it is nowhere described in the text or figures.

4. Minor: Figure 1c right panel could enumerate both the fitted values based on the model in white but also include the experimental values indicated by the bar chart mean (maybe in black) for ease of comparison. I assume the fitted values are placed in the figure to reiterate the point that the model is predicting the experimental well in this distribution of fluorescently labelled H2A, so having both the numbers side by side would better home in on that point. Moreover, if one does not read the text, it could be easily misinterpreted that the white values are the experimental values from the chart.

5. Minor: The use of the phrase “stoichiometry distribution” in line 131 and Figure 1c unclear or uncommon. It would be nice if it was explicitly defined. I understand what stoichiometry means and what a distribution means, but I do not know of a commonly accepted definition of a stoichiometry distribution although it can be inferred from the text.

6. Minor: Supplementary Figure 1.1.a uses abbreviations (in b/w). I assume it meant in black and white. Not sure how common this abbreviation is, but they could define for the first time in Figure 1.1.a and use hence forth like in Figure 1.1.b.

7. Minor: Supplementary Figure 1.1.b is a nice gel with a good explanation of the experiment. However, enumerating the lanes at the bottom of the gel would make it easier to reference the last experimental condition in the text where MCM2-7 only loads in the presence of ATP but not ATPγS.

8. Medium: In supplementary Figure 1.1.c please label that both the left and right panels are dCas9 lifetime analyses with different fluorophores on the plots themselves. At first glance one cannot tell if the plots are comparing the labelled ORCs, MCMs, histones or dCas9 proteins from the coloured gels and annotation in the above panels. The plots themselves need to mention what they are measuring. Moreover, how was supplementary Figure 1.1.c experimentally set up? It is not clear if the lifetime analysis of the fluorophores were done in bulk with a cuvette like set up or done in the Q-trap on DNA, from the methods text it seems to be done in the Q-trap. Maybe having a schematic of the experiment or better explaining how the survival probability is determined in the supplementary figures’ text will make it clearer to readers how the plots were determined. Because the plot is placed right under the gel of a bulk experiment, it feels like the lifetime analysis plot was from a bulk experiment. For example, “single molecule lifetime analysis of fluorophores...” or “single molecule photobleaching control” in the figures text would make it clearer. It was a little confusing to the reader when suddenly Supplementary Figure 1.1.c was being compared to Supplementary Figure 1.3.c. and the plots were represented slightly differently. Supplementary Figure 1.1.c has a longer x-axis for time and no title but, Supplementary Figure 1.3.c, the representation of the same type of data, gets titled and a shorter plot. This is trivial but it would be nice if more consistent.

9. Medium/Major: Supplementary Figure 1.3 was difficult to dissect.

a. Panel a of this figure needs to be redrawn, scaled evenly, and/or labelled more clearly. The colour scheme and cylinder representations of the nucleosome and its representative parts are almost too simplified. In part 1 of panel a, the nucleosome almost looks like the NPS sites on the DNA. And then when the nucleosome is drawn in panel 2 it is suddenly larger and is being broken down into smaller cylinders. When reading the text one can figure out what cylinder represents what, but it would be so

much easier on the readers if the parts were clearly labelled in the schematic. The smaller cylinder is H2A/H2B, the larger middle cylinder is the H3/H4 hexasome, when the hexasome is bound to one H2A/H2B it is a tetrasome. When all together it is the nucleosome octamer. Therefore, when you get to part 3 it makes sense when you bring in the ellipse as a representative label, and why it is bound to the H2A/H2B smaller cylinder. The colour scheme and the small white text in the ellipse as AF488 is extremely difficult to read especially when printed on standard A4.

b. Panel b of this figure could be explained a bit more. Why you decided to show this diagram could be made more explicit, else I feel it confuses the reader into thinking your model is a terrible predictor of the experimental overall at first glance and into questioning why you even modelled the data in the first place. Moreover, maybe make it more clear how the experimental data was determined to compare against the model. I assumed it was done via photobleaching the foci. The red dot to indicate where your model predicts the experimental best is not very visible against the purple, especially when printed. Maybe this can be avoided by changing the colour scheme or using a white dot? With further deliberation maybe this plot is clearer in Figure 1.c, so it is parallel to Supplementary Figure 1.4.

c. Panel c is a plot of fluorophore lifetime analysis of H2A via photobleaching on DNA containing an origin. Whilst the main text references it as a photobleaching approach, maybe just say, "Lifetime analysis of H2A via photobleaching." in the supplementary legend itself to make it easier for the reader.

10. Medium: Supplementary Figure 1.4 centre panel of the stoichiometry distribution is even more confusing than Figure 1.c. I do not get the point of overlaying a barely visible grey bar graph on the purple bars. Why not have the predicted distribution and then experimental distribution side by side? Also why only have the white numbers inside the bars designating the fitted values based on the model, when you can list both experimental and predicted numbers or none with the having the different colour bars side by side so its explicitly visible and comparable? The right most panels with landscapes of the mean squared error between fit and experiment demonstrated here are not clear. It helps compare the different sample preparations and it is convincing the Sample 1 to 3 look similar, but Sample 4 looks different and more badly predicted by the model. Maybe this data could help convince the reader that most of your experimental foci have at least two NPS sites occupied and therefore your origin sequences are indeed flanked by 2 nucleosomes/hexasomes. Again, the red dot in the right panels is not distinctly visible from the purple contour colours. Also, what is this PDF on the left y-axis of the left panels?

11. Minor: Figure 2.a shows a schematic of the experimental design. It seems the opaque red cartoons are representations of ORC. While the text gives details of rapid ORC binding experiments, the figure legend just describes it as incubation method pertaining to the panels below. It would be nice if every cartoon representation is labelled in the first instance of being shown.

12. Minor: Figure 2 onwards is encapsulated in a large grey box that seems unnecessary and unattractive. Figure 1 and some of the supplementary figures were not given a large grey box – why the inconsistency?

13. Major: Figure 2, and all the following figures representing experiments, never show representative kymographs of any of their experimental set ups.

a. In Figure 2.b the plots are quantifying the spatial distribution of fluorescent ORC on the DNA from the centre of the DNA for different DNA substrates. Whilst a kymograph of this data should only show some foci in between the beads, it would be reassuring to see an example kymograph of each condition in the supplementary or above each plot with its cartoon representation. The beauty of single molecule experiments is their directly visual nature and directly showing a kymograph will make it easier for readers to dissect what the plots represent. Also, what is this PDF on the left y-axis again?

b. In Figure 2.b the experiments show the spatial distribution of rapid binding ORC to variety of DNA. On the chromatinized origins, are the experimenters using nucleosomes-bound AF488-H2A (aka labelled histones)? From section 1 of the results titled, "Establishing a labelled chromatinized origin of replication within 10.4 kbp DNA for single molecule investigations," the text and the use of the same cartoon schematics from Figure 1 makes the readers assume that all the chromatinized DNA in this established assay has labelled histones. If this is the case, why not show the representative kymographs of ORC localizing at the site of the nucleosome and origins? Such an experiment and graph would be the most convincing evidence of their co-localization and ORC preferentially interacting with the chromatinized origins. However, if this experiment does not use labelled histones, they should be more explicit that this is not the case and that they solely determine the localization of ORC at origins via a predetermined function of genomic coordinates. Moreover, if they did not use labelled histones, why did they not use it? It seems an easy enough experiment for them to repeat with labelled histones, and if there is no difference between the two it would be even more convincing evidence that their labelling strategy does not affect the nucleosomes' structure, function, and interactions with ORC. The Q-trap is capable of multiple colour experiments, the researchers should be able to make use of it.

c. Why did the researchers decide that foci containing more than 5 ORCs were aggregates and were not analyzed further? Supplementary Figure 2 quantifies that the four different DNA conditions mostly had foci only containing 1 ORC. But were there a lot of foci containing more than 5 ORCs/aggregating? How would the data change with an analysis of looking at foci with only 1-5 ORCs versus 1-6 ORCs versus 1-10 ORCs? An explanation would be nice, because 4 ORCs could also potentially represent an aggregate. If there was an explanation of the chosen cut-off, like only a certain number of ORCs can bind in that diffraction limited spot or that this is the conventional cut off for aggregates, it would make the chosen cut-off feel less arbitrary and trust that most of the relevant data does only contain 1 ORC.

14. Medium: Figure 3 looking at the mobility of ORC on said chromatinized DNA

a. It would be nice to see at least one representative kymograph with the overlay of the traces illustrating the motion of JF646 ORC for the different experimental conditions.

b. Figure 3.a.i and ii shows 30% of all traces selected at random or ORCs starting in the chromatinized origin bin or not respectively. Why not show all the traces in the figure or in the supplementary? And why not show a cartoon schematic of the Q-trap beads/chromatinized DNA/ORC on the y-axis so it will be clearer that the traces in this specific figure represent ORC mobility only on chromatinized DNA. It is convincing that the ORCs move more when not starting in the chromatinized origin.

c. Again, are these experiments done with labelled histones or not? If so representative kymographs for Figure 3.a.i and ii would be even more compelling to see.

d. Figure 3b seems to quantify the motion of ORC in the above traces, but for all four experimental conditions. Why only label the axis on the outer edges of the subpanels? It would be easier for readers to dissect the diffusion constants if the axes for each subpanel were labelled.

e. Figure 3c subpanels then look at slowly moving or static ORC population. Whilst $D < D^*$ is meant to represent the 'slow ORC population' as indicated in text, it would be easier if the graphs were titled/labelled with 'slow ORC population'. Readers can then directly understand you have a slower ORC population with ARS or chromatinization without having to read or reference the text.

15. Minor/Medium: Supplementary Figure 3.b

a. Instead of labelling subplots with $D < D^*$ indicating the slow ORC population and then $D > D^*$ indicating the fast or diffusive ORC population it would be clearer to also label the panels with "slow or fast ORC population". A little confused on why the slow ORC lifetime was drastically higher with the

mutated chromatinized origins, as seen in left subpanel of Supplementary Figure 3.b. iv. An explanation would be helpful.

b. It would be nice if figure legend explained that survival probability could mean a mixture of photobleaching or dissociation of protein from DNA, even though the text references Supplementary Figure 1.1.c to suggest its more likely dissociation, an almost equally substantial portion can still photo bleach within 20-30 seconds according to the same supplementary figure. The legend and main text almost imply survival probability is equal to dissociation of these different population of ORCs from DNA. The term survival probability is used multiple times in the figure legend and having it explicitly defined in the first instance would be nice, so readers are not forced to infer what it means after reading the main text for clarification.

16. Major: Figure 4 and Supplementary Figures 4s

a. Why did the researchers perform the extended incubation in bulk for 30 minutes in an Eppendorf as depicted in Figure 4a and Supplementary Figure 4a and not in the Q-trap protein channel? How would this change the experimental results? Now the experiments prior do not seem directly comparable. Are you now preferentially catching/pulling on different ORC populations with different lifetimes on the DNA? Are you preferentially tethering certain DNA populations?

b. Figure 4b.iv needs to be repeated so that there are more than 2 DNA molecules analyzed when the other conditions have substantially more experimental samples. I do not think it can be compared to the other conditions with such low numbers.

c. Supplementary Figure 4.1 shows us the same analysis of labelled histones using this near incubation condition in the tube before transferring to the Q-trap. The comments made for supplementary Figure 1.4 all apply.

d. Supplementary Figure 4.1 makes it seem like the experiments in Figure 4 are done with labelled histones, but that is not clear. Again, are all these experiments done with labelled histones? In Supplementary Figure 4a the cartoon schematic representing histones labelled histones are used in panel b of Figure 4.

e. Supplementary Figure 4.2 could be depicted more clearly. Similar comments about the axes from above. A concerning aspect of the representation is that the split between "all" foci analyzed on the left graph versus the "in ori bin" foci analyzed on the right graph of each subpanel. Readers must do the maths themselves to interpret the data or have explicitly read the text to get the percentages or ORC bound to origins versus overall on the DNA. Also, is it worth representing the data in subpanel iii and iv in the same way for the DNA you can see multiple foci for? I suggest avoiding to show the statistics of $N_{\text{foci}} = 1$ or 0 with error bars coming off the graph or nothing on the graph. Maybe put the NDNA-no ORC on the plots too, else at first glance it looks like you caught no DNA and saw no foci for the mutated origin conditions and lose the main point that you get little ORC stably bound on the mutated DNA with this experimental set up.

f. Supplementary Figure 4.3 looking at the stability of labelled chromatin with the bulk ORC and Cdc6 incubation can have the same comments that apply to Supplementary Figure 1.4. It is interesting to see how adding different experimental components like Cdc6 changes how well H2A dimers occupy foci and therefore how well model predicts the experimental. Do the extended incubations directly show the presence of ORC and Cdc6 affects nucleosome positioning? I would be more convinced if the incubations were tracked in the Q-trap like experiment of Figure 3.

17. Medium: Figure 5 and Supplementary Figure 5s

a. In Figure 5a, the first representative cartoon of MCMs should be labelled. Horizontal opaque red blobs are not massively distinct from vertical opaque red blobs.

- b. Again explaining why design the reaction to happen in a tube before introducing into the flow cell and not do the reaction in the Q-trap for 30 minutes before tethering to beads would be useful? Or why not tether DNA and then introduce to protein channel in the Q-trap? It would be nice to have both experiments and/or an explanation for this chosen experimental set up.
- c. Are these experiments conducted with labelled histones? If so, make clearer, if not, why not?
- d. Why no representative kymographs of the labelled MCMs on DNA? Examples of MCM localizing at the ARS sites with nucleosomes can be included.
- e. Supplementary Figure 5.1, again why make reader do the maths themselves to interpret the data of the fraction of MCMs bound to origin bin increases with ARS and nucleosomes compared to the other conditions?
- f. Supplementary Figure 5.2 have the same comments that apply to Supplementary Figure 1.4.

Reviewer #3 (Remarks to the Author):

The manuscript by Sánchez and colleagues investigated how the presence of origin-flanking nucleosomes affects the binding and dynamics of ORC proteins and MCM loading on the origin recognition sequences, mainly by using the *in vitro* single-molecule optical tweezer coupled with fluorescence assay. They found that the presence of nucleosomes surrounding origins provides a mild increase in local binding but a reduction in mobility of ORC, and contributes to the loading and spatial constraint of MCM. In addition, they investigated the influence of ORC and MCM on the nucleosomes around the origin, and found ORC can broaden the spatial distribution and reduce the stoichiometry of H2A in the nucleosomes and observed a protective role for MCM in the maintenance of nucleosome. By using the purified system, this study directly provides new insights into the important question of how the local nucleosomes function in the origin licensing during DNA replication in yeast, and also provides a good platform to further explore the mechanism and the epigenetic regulation of chromatin replication processes.

Overall, the experiments largely addressed the effect of origin-flanking nucleosomes on the binding and dynamics of ORC proteins and MCM. However, there are a few questions that should be clarified, as described in details below:

1. It would be helpful to give more details on the DNA construct used. As for the Figure 1b up right panel, the author should give more details on the center fragment of the DNA template, including to clarify the length of the ARS1, and the distance between the ARS1 and the flanking NPS.
2. Does the distance between the ARS1 and the flanking nucleosomes affect the results, for example, the binding of ORC and MCM on ARS1? The author should try different lengths of the distance to clarify how it affects and clarify why they choose this distance in the study.
3. In the Figure 1b right lower panel, why only one single spot is detected in the middle? it should see two spots for the two nucleosomes located at the two NPSs?
4. For the Figure 2, 4 and 5, the typical confocal scans for the signal of fluorescent-labeled ORC, MCM or the nucleosomes should be presented for the four DNA and chromatin templates.
5. Only observed the change of fluorescence labeled-H2A cannot indicate the change of NPS site occupancy. It would be helpful to observe the change of fluorescence labeled-H3 or H4 to confirm these results.
6. We know a histone octamer wrapped more than 140 bp DNA to form a nucleosome. To unwrap one intact nucleosome, the length change of DNA should be around 45 nm, which has been shown by several force spectroscopy studies on the nucleosome unwrapping process. But in this work, the author claimed that they observed one jump at a force of 17.6 pN, with the length increments of 24.8 nm for the nucleosome force spectroscopy. It looks the nucleosomes used in this study were not well reconstituted, with only sub-nucleosomes or tetrasomes are formed.
7. It should be interested to observe, how the binding of ORC or MCM affect the property, including the stability or unfolding kinetics of the flanking nucleosomes?

Delft, August 31, 2023

We thank the reviewers of our manuscript for their interest in the work and positive endorsement. They have clearly studied the manuscript with interest, and we thank them for their comments, questions, and recommendations. Below, we provide a point-by-point response (in blue) to their questions (in black italics), and we have indicated where modifications to either the main text or the supplementary information have been made. In the manuscript itself, changes are highlighted in yellow.

Kind regards,

The authors

REVIEWER COMMENTS

Reviewer #1 (Remarks to the Author):

The manuscript of Sánchez et al., uses an in vitro single-molecule approach to study a chromatinized origin of replication. They find that ORC binds the origin with similar efficiency, whether chromatinized or not, but its mobility is reduced by nucleosomes. MCM recruitment on a chromatinized origin is shown to be efficient, but its subsequent movement away from the origin is limited. Together these results suggest that chromatinized origins in yeast are crucial for local MCM retention and thus might facilitate replisome assembly.

The study is very well executed and clear. The consistent experiments with the four different DNA constructs with/out the specific binding site and with/out the nucleosomes present give a very clear data set that in turn lead to significant conclusions. They leverage the single molecule instrumentation in a smart way and provide nice statistic with their data to underpin their conclusions. All in all, it makes me very enthusiastic about their results and I think that the manuscript is very suitable for Nature communication.

Response: We thank the referee for this encouraging endorsement!

I have only a few minor comments that should be addressed:

-The binning choice for the localization of the proteins on the DNA is not clear to me. They choose it to be the diffraction limit but there is no need to link it to that because the localization precision can be better than the diffraction limit.

-So that leads to the question what really limits the size of the bin in these experiments?

-And is it possible to reduce the bin size in order to get more spatial resolution?

Response: We thank the referee for these reasonable questions. Under our imaging conditions, the localization error for a single fluorophore is approximately 100 bp (as determined with dCas9^{AF488} and dCas9^{JF646}), which gives us the relative uncertainty in position. But, we also have an absolute uncertainty in determining the exact position of a fluorophore on the DNA, which relates to the uncertainty as to where the DNA extremity is located (which can also be determined with dCas9^{AF488} and dCas9^{JF646}). This is taken into account in our spatial distribution plots whose x-axes are genomic location. Because we in general find that these uncertainties increase when we work with proteins whose binding is more complex binding than that of dCas9, we ultimately select a conservative bin size (of 670 bp) that encompass the (chromatinized) origin into a single bin. That this happens to be close to the diffraction limit is a bit by chance, the referee is of course correct that the diffraction limit itself does not apply to localization precision.

We have included this clarification in the Methods section (lines 881-887).

Reviewer #2 (Remarks to the Author):

Review of "A chromatinized origin reduces the mobility of ORC and MCM through interactions and spatial constraint"

This work from the Dekker lab investigates the role of nucleosomes flanking origin sequences on ORC and MCM binding as well as their interactions with the nucleosomes using a single molecule technique. Overall, the data does back up the claims that nucleosomes can facilitate ORC binding at the origin and decrease ORC mobility. The data also shows that

the nucleosomes spatially constrain MCMs that may be loaded at the origin. It was also nice that the authors controlled for the effect of the origin sequence or nucleosomes in combination.

Response: We thank the reviewer for appreciating our experimental design and acquired datasets.

While the text is well written, the figures should be improved upon. Figures were often not very clear, could not stand alone, and/or sometimes did not directly show the main point referenced in the text.

Response: We had made quite an effort to produce clear Figures, but we thank the referee for the time spent in studying them closely and suggesting improvements. We have modified them as described below.

Another major issue was the lack of any representative kymographs for any of the experimental conditions except for the development of the assay.

Response: Following the referee's suggestions and those of referee 3, we have included representative images and kymographs as supplementary figures, as described below.

It was also unclear if the exact assay, using labelled histones described in the first part of the results, was consistently used throughout the paper and why the authors potentially chose different experimental set ups.

Response: Indeed, we have consistently used the labeled histones throughout this study. We now emphasize this point more clearly in the text, as described below.

The discussions on the implications of chromatinized DNA favouring CMG formation and the initiation of DNA replication at sequence specific origins is a promising hypothesis.

Response: We are very happy that the referee has engaged with us on this hypothesis!

Revision points as they came up in text or figures:

1. Minor: Use of the word 'attendant' as an adjective in abstract (line 22) and in the introduction (line 40) is confusing. Whilst it is used correctly, it is not a commonly used adjective or often used in adjective form and can be more easily understood with synonyms like resulting or consequent. In line 22 it could even be omitted.

Response: We have retained this word in line 22 (where we think it works very well), but have removed it in line 40.

2. Minor: What were the known standards for determining labelling efficiency for histones to determine degree of labelling to be $plabel = 0.81$ (Line 83)? In Line 84 the reference to methods and statement: as determined from bulk experiments is not clear. Or maybe more details in methods on how to determine labelling efficiency – how do they determine the protein concentrations and fluorophore concentrations against said standards?

Response: We have included additional details in the methods section (lines 707-709 and lines 718-720) to explain that the total protein concentration is determined using BSA as a protein standard and that the labelled protein concentrations are determined spectrophotometrically by measuring the absorbance at 488 nm (for AF488 dye) or at 646 nm (for JF646 dye).

3. Medium: Figure 1c left panel makes no explanation of the right y-axis. What is this "PDF (1/kbp)"? I assume it some probability density function, but it is nowhere described in the text or figures.

Response: We thank the referee for pointing out this omission. Indeed, PDF is the probability density function. We have expanded the information in all relevant figure captions.

4. Minor: Figure 1c right panel could enumerate both the fitted values based on the model in white but also include the experimental values indicated by the bar chart mean (maybe in black) for ease of comparison. I assume the fitted values are placed in the figure to reiterate the point that the model is predicting the experimental well in this distribution of fluorescently labelled H2A, so having both the numbers side by side would better home in on that point. Moreover, if one does not read the text, it could be easily misinterpreted that the white values are the experimental values from the chart.

Response: Indeed, the intent of the numbers in white is to show the fit values in the context of a bar plot. We think that adding more numbers will make the figure less clear; instead, we have simplified the figure by replacing the white numbers inside the bars with a filled white circle. We think this makes the point more clearly.

5. Minor: The use of the phrase “stoichiometry distribution” in line 131 and Figure 1c unclear or uncommon. It would be nice if it was explicitly defined. I understand what stoichiometry means and what a distribution means, but I do not know of a commonly accepted definition of a stoichiometry distribution although it can be inferred from the text.

Response: We do not readily see how this can be unclear. Distributions can plot any quantity, in this case it is stoichiometry. Therefore it is reasonable to call this a stoichiometry distribution.

6. Minor: Supplementary Figure 1.1.a uses abbreviations (in b/w). I assume it meant in black and white. Not sure how common this abbreviation is, but they could define for the first time in Figure 1.1.a and use hence forth like in Figure 1.1.b.

Response: Indeed b/w stands for black and white. We have clarified the figure caption.

7. Minor: Supplementary Figure 1.1.b is a nice gel with a good explanation of the experiment. However, enumerating the lanes at the bottom of the gel would make it easier to reference the last experimental condition in the text where MCM2-7 only loads in the presence of ATP but not ATPgS.

Response: We have enumerated the lanes at the bottom and clarified the figure caption.

8. Medium: In supplementary Figure 1.1.c please label that both the left and right panels are dCas9 lifetime analyses with different fluorophores on the plots themselves. At first glance one cannot tell if the plots are comparing the labelled ORCs, MCMs, histones or dCas9 proteins from the coloured gels and annotation in the above panels.

Response: We have labeled the figure as suggested.

The plots themselves need to mention what they are measuring. Moreover, how was supplementary Figure 1.1.c experimentally set up? It is not clear if the lifetime analysis of the fluorophores were done in bulk with a cuvette like set up or done in the Q-trap on DNA, from the methods text it seems to be done in the Q-trap. Maybe having a schematic of the experiment or better explaining how the survival probability is determined in the supplementary figures' text will make it clearer to readers how the plots were determined. Because the plot is placed right under the gel of a bulk experiment, it feels like the lifetime analysis plot was from a bulk experiment. For example, “single molecule lifetime analysis of fluorophores...” or “single molecule photobleaching control” in the figures text would make it clearer. It was a little confusing to the reader when suddenly Supplementary Figure 1.1.c was being compared to Supplementary Figure 1.3.c. and the plots were represented slightly differently. Supplementary Figure 1.1.c has a longer x-axis for time and no title but, Supplementary Figure 1.3.c, the representation of the same type of data, gets titled and a shorter plot. This is trivial but it would be nice if more consistent.

Response: We have modified the caption of Supplementary Figure 1.1c as suggested. Similar changes have been made to Supplementary Figure 1.3c and Supplementary Figure 3.2.

9. Medium/Major: Supplementary Figure 1.3 was difficult to dissect.

a. Panel a of this figure needs to be redrawn, scaled evenly, and/or labelled more clearly. The colour scheme and cylinder representations of the nucleosome and its representative parts are almost too simplified. In part 1 of panel a, the nucleosome almost looks like the NPS sites on the DNA. And then when the nucleosome is drawn in panel 2 it is suddenly larger and is being broken down into smaller cylinders. When reading the text one can figure out what cylinder represents what, but it would be so much easier on the readers if the parts were clearly labelled in the schematic.

The smaller cylinder is H2A/H2B, the larger middle cylinder is the H3/H4 hexasome, when the hexasome is bound to one H2A/H2B it is a tetrasome. When all together it is the nucleosome octamer. Therefore, when you get to part 3 it makes sense when you bring in the ellipse as a representative label, and why it is bound to the H2A/H2B smaller cylinder. The colour scheme and the small white text in the ellipse as AF488 is extremely difficult to read especially when printed on standard A4.

Response: Following the referee's suggestions, we have modified the figure.

b. Panel b of this figure could be explained a bit more. Why you decided to show this diagram could be made more explicit, else I feel it confuses the reader into thinking your model is a terrible predictor of the experimental overall at first glance and into questioning why you even modelled the data in the first place.

Response: The plot shows that while other pairwise combinations of p_{H2A} and $p_{occupancy}$ are possible, the associated errors will be higher than those of the optimal solution: most of these combinations will fall in the light blue area corresponding to poor fits. The pair of values associated with the minimum error (now indicated by a white dot, see a few lines down) is the best descriptor of the experimental values.

Moreover, maybe make it more clear how the experimental data was determined to compare against the model. I assumed it was done via photobleaching the foci.

Response: The experimental data was determined via photobleaching, as indicated in the main text (line 124).

The red dot to indicate where your model predicts the experimental best is not very visible against the purple, especially when printed. Maybe this can be avoided by changing the colour scheme or using a white dot?

Response: We have changed the color of this dot to white, as requested.

With further deliberation maybe this plot is clearer in Figure 1.c, so it is parallel to Supplementary Figure 1.4.

Response: We have kept the error landscape in the supplement (as we do in Supplementary Figure 1.4) to maintain the focus of the main text.

c. Panel c is a plot of fluorophore lifetime analysis of H2A via photobleaching on DNA containing an origin. Whilst the main text references it as a photobleaching approach, maybe just say, "Lifetime analysis of H2A via photobleaching." in the supplementary legend itself to make it easier for the reader.

Response: See our response under point 8.

10. Medium: Supplementary Figure 1.4 centre panel of the stoichiometry distribution is even more confusing than Figure 1.c. I do not get the point of overlaying a barely visible grey bar graph on the purple bars. Why not have the predicted distribution and then experimental distribution side by side? Also why only have the white numbers inside the bars designating the fitted values based on the model, when you can list both experimental and predicted numbers or none with the having the different colour bars side by side so its explicitly visible and comparable? The right most panels with landscapes of the mean squared error between fit and experiment demonstrated here are not clear.

It helps compare the different sample preparations and it is convincing the Sample 1 to 3 look similar, but Sample 4 looks different and more badly predicted by the model. Maybe this data could help convince the reader that most of your experimental foci have at least two NPS sites occupied and therefore your origin sequences are indeed flanked by 2 nucleosomes/hexasomes. Again, the red dot in the right panels is not distinctly visible from the purple contour colours. Also, what is this PDF on the left y-axis of the left panels?.

Response: See our responses under points 3 and 4. We have also removed the superimposed grey bars from Figure 1c, Supplementary Figure 1.4, Supplementary Figure 4.1, Supplementary Figure 4.3, and Supplementary Figure 5.2.

11. Minor: Figure 2.a shows a schematic of the experimental design. It seems the opaque red cartoons are representations of ORC. While the text gives details of rapid ORC binding experiments, the figure legend just describes it as incubation method pertaining to the panels below. It would be nice if every cartoon representation is labelled in the first instance of being shown.

Response: We thank the referee for this suggestion and have included the labeling as requested.

12. Minor: Figure 2 onwards is encapsulated in a large grey box that seems unnecessary and unattractive. Figure 1 and some of the supplementary figures were not given a large grey box – why the inconsistency?.

Response: We had meant the differently shaded boxes to indicate different incubation conditions, but after some deliberation we agree with the referee that they can be removed. We have adapted the caption of Figure 3 (the only one without a schematic) that the data refer to the incubation conditions shown in Figure 2a.

13. Major: Figure 2, and all the following figures representing experiments, never show representative kymographs of any of their experimental set ups.

a. In Figure 2.b the plots are quantifying the spatial distribution of fluorescent ORC on the DNA from the centre of the DNA for different DNA substrates. Whilst a kymograph of this data should only show some foci in between the beads, it would be reassuring to see an example kymograph of each condition in the supplementary or above each plot with its cartoon representation. The beauty of single molecule experiments is their directly visual nature and directly showing a kymograph will make it easier for readers to dissect what the plots represent.

Response: A kymograph creates a visual representation of a dynamic process in a single image by stacking up the intensities measured for a given line scan in the confocal scanning area at consecutive time intervals. In Figure 2b, we do not show the dynamics of ORC, but rather a snapshot of its initial position. Therefore, to accommodate the referee's request, we have included typical confocal scans, in the color channels analyzed: red for ORC and blue for H2A, along with the histograms representative of each experimental scenario below the schematics and above the histograms in Figure 2b.

Also, what is this PDF on the left y-axis again?.

Response: See our response under point 3.

b. In Figure 2.b the experiments show the spatial distribution of rapid binding ORC to variety of DNA. On the chromatinized origins, are the experimenters using nucleosomes-bound AF488-H2A (aka labelled histones)? From section 1 of the results titled, "Establishing a labelled chromatinized origin of replication within 10.4 kbp DNA for single molecule investigations," the text and the use of the same cartoon schematics from Figure 1 makes the readers assume that all the chromatinized DNA in this established assay has labelled histones. If this is the case, why not show the representative kymographs of ORC localizing at the site of the nucleosome and origins?

Such an experiment and graph would be the most convincing evidence of their co-localization and ORC preferentially interacting with the chromatinized origins. However, if this experiment does not use labelled histones, they should be more explicit that this is not the case and that they solely determine the localization of ORC at origins via a predetermined function of genomic coordinates.

Moreover, if they did not use labelled histones, why did they not use it? It seems an easy enough experiment for them to repeat with labelled histones, and if there is no difference between the two it would be even more convincing evidence that their labelling strategy does not affect the nucleosomes' structure, function, and interactions with ORC. The Q-trap is capable of multiple colour experiments, the researchers should be able to make use of it.

Response: All chromatin experiments were performed with labeled nucleosomes. We have clarified this by replacing 'H2A' with 'H2A^{AF488}' and 'nucleosomes' with 'labelled nucleosomes' throughout the experimental descriptions in the manuscript, including figure captions when referring to labeled histones or nucleosomes. See our responses under point 13.a.

c. Why did the researchers decide that foci containing more than 5 ORCs were aggregates and were not analyzed further? Supplementary Figure 2 quantifies that the four different DNA conditions mostly had foci only containing 1 ORC. But were there a lot of foci containing more than 5 ORCs/aggregating?

How would the data change with an analysis of looking at foci with only 1-5 ORCs versus 1-6 ORCs versus 1-10 ORCs? An explanation would be nice, because 4 ORCs could also potentially represent an aggregate. If there was an explanation of the chosen cut-off, like only a certain number of ORCs can bind in that diffraction limited spot or that this is the conventional cut off for aggregates, it would make the chosen cut-off feel less arbitrary and trust that most of the relevant data does only contain 1 ORC

Given the footprint of ORC complex on the DNA (~ 50 bp) (Li, et al. Nature 2018), up to 15 ORCs could bind DNA within a single diffraction-limited spot of ~750 bp. To avoid counting ORC aggregates, we employ an apparently conservative cut-off of 5. Still, while the percentage of foci with more than 5 ORCs varies somewhat per experimental conditions (Sanchez, et al. Nat. Comm. 2021), it is never very high: 7% (11 out of 156 total foci) for DNA molecules containing the non-chromatinized ARS1 origin; 4% (3 out of 70 total foci) for DNA molecules containing the chromatinized ARS1 origin; 18% (22 out of 122 total foci) for DNA molecules containing the non-chromatinized mutated origin; and 3% (2 out of 72 total foci) for DNA molecules containing the chromatinized mutated origin. Therefore, whether we changed this value to e.g. a somewhat higher value would not influence our overall findings.

14. Medium: Figure 3 looking at the mobility of ORC on said chromatinized DNA

a. It would be nice to see at least one representative kymograph with the overlay of the traces illustrating the motion of JF646 ORC for the different experimental conditions.

Response: We have included a new supplementary Figure 3.1 that shows representative kymographs, as requested.

b. Figure 3.a.i and ii shows 30% of all traces selected at random or ORCs starting in the chromatinized origin bin or not respectively. Why not show all the traces in the figure or in the supplementary?

Response: We have chosen to show a fraction (30%) of all the traces to make them visible, as showing them all would make the plot too crowded.

And why not show a cartoon schematic of the Q-trap beads/chromatinized DNA/ORC on the y-axis so it will be clearer that the traces in this specific figure represent ORC mobility only on chromatinized DNA. It is convincing that the ORCs move more when not starting in the chromatinized origin.

Response: Introducing more graphical elements into the figure makes it difficult to read. We believe that the figure is clearer without additional cartoons, as labels and dotted lines already indicate the experimental conditions used.

c. Again, are these experiments done with labelled histones or not? If so representative kymographs for Figure 3.a.i and ii would be even more compelling to see.

Response: All chromatin experiments were performed with labelled nucleosomes. See our responses under point 13b and 14.a.

d. Figure 3b seems to quantify the motion of ORC in the above traces, but for all four experimental conditions. Why only label the axis on the outer edges of the subpanels? It would be easier for readers to dissect the diffusion constants if the axes for each subpanel were labelled.

e. Figure 3c subpanels then look at slowly moving or static ORC population. Whilst $D < D^$ is meant to represent the 'slow ORC population' as indicated in text, it would be easier if the graphs were titled/labelled with 'slow ORC population'. Readers can then directly understand you have a slower ORC population with ARS or chromatinization without having to read or reference the text.*

Response: We consider that these last two suggestions from the referee are a matter of taste. We opt for a minimalist display that still conveys all the information.

15. Minor/Medium: Supplementary Figure 3.b

a. Instead of labelling subplots with $D < D^$ indicating the slow ORC population and then $D > D^*$ indicating the fast or diffusive ORC population it would be clearer to also label the panels with "slow or fast ORC population". A little confused on why the slow ORC lifetime was drastically higher with the mutated chromatinized origins, as seen in left subpanel of Supplementary Figure 3.b. iv. An explanation would be helpful.*

Response: Regarding panel labelling, we refer to our previous response 14.b. Regarding the higher ORC lifetime obtained for DNA molecules containing the chromatinized mutated origin, we speculate that this could be an effect of a lower number of data points acquired compared to on DNA molecules containing the chromatinized ARS1 origins. Overall, however, the average lifetime of the slow ORC population is typically shorter than that of the fast ORC population.

b. It would be nice if figure legend explained that survival probability could mean a mixture of photobleaching or dissociation of protein from DNA, even though the text references Supplementary Figure 1.1.c to suggest its more likely dissociation, an almost equally substantial portion can still photo bleach within 20-30 seconds according to the same supplementary figure. The legend and main text almost imply survival probability is equal to dissociation of these different population of ORCs from DNA. The term survival probability is used multiple times in the figure legend and having it explicitly defined in the first instance would be nice, so readers are not forced to infer what it means after reading the main text for clarification

Response: We have included the definition of survival probability as the fraction of fluorophores visible at a given time in the caption of Supplementary Figure 1.1.c. See also our response under point 8.

16. Major: Figure 4 and Supplementary Figures 4s

a. Why did the researchers perform the extended incubation in bulk for 30 minutes in an Eppendorf as depicted in Figure 4a and Supplementary Figure 4a and not in the Q-trap protein channel? How would this change the experimental results?

Now the experiments prior do not seem directly comparable. Are you now preferentially catching/pulling on different ORC populations with different lifetimes on the DNA? Are you preferentially tethering certain DNA populations?

Response: We performed the extended incubation in bulk for 30 min in an Eppendorf tube because we wanted to compare it with the experiments in Figure 5 that show MCM loading. Efficient loading of MCM requires this relatively long incubation time (Remus et al., Cell 2009; Miller et al., Nature 2019; Sanchez et al., Nat. Comm. 2021). By splitting the experiments into two parts (bulk and single molecule) we increase the efficiency of loading and the yield of molecules that can be analyzed in the Q-trap.

b. Figure 4b.iv needs to be repeated so that there are more than 2 DNA molecules analyzed when the other conditions have substantially more experimental samples. I do not think it can be compared to the other conditions with such low numbers.

Response: The message of the figure is precisely to show that after incubation in bulk, there is hardly any DNA molecule (2 out of a total of 47) that retains ORC, as we explained in the main text from line 288 onwards. Based on the referee's comment, we have modified the caption to include the total number of DNA molecules analyzed.

c. Supplementary Figure 4.1 shows us the same analysis of labelled histones using this near incubation condition in the tube before transferring to the Q-trap. The comments made for supplementary Figure 1.4 all apply.

Response: We have changed Supplementary Figure 4.1 accordingly. See our responses under points 3, 4, and 10.

d. Supplementary Figure 4.1 makes it seem like the experiments in Figure 4 are done with labelled histones, but that is not clear. Again, are all these experiments done with labelled histones? In Supplementary Figure 4a the cartoon schematic representing histones labelled histones are used in panel b of Figure 4.

Response: See our responses under points 13 and 14.

e. Supplementary Figure 4.2 could be depicted more clearly. Similar comments about the axes from above. A concerning aspect of the representation is that the split between "all" foci analyzed on the left graph versus the "in ori bin" foci analyzed on the right graph of each subpanel. Readers must do the maths themselves to interpret the data or have explicitly read the text to get the percentages or ORC bound to origins versus overall on the DNA. Also, is it worth representing the data in subpanel iii and iv in the same way for the DNA you can see multiple foci for? I suggest avoiding to show the statistics of $N_{\text{foci}} = 1$ or 0 with error bars coming off the graph or nothing on the graph. Maybe put the NDNA-no ORC on the plots too, else at first glance it looks like you caught no DNA and saw no foci for the mutated origin conditions and lose the main point that you get little ORC stably bound on the mutated DNA with this experimental set up.

Response: Based on the referee's suggestion, we have included the percentage of ORC bound to DNA in the origin bin in the plots. And regarding the question about panels iii and iv in Supplementary Figure 4.2, we agree with the referee that these statistics are low but we decided that it was better to include these panels for consistency with the other figures

in the manuscript and supplement. The low number of data points in these panels is simply due to the fact that ORC does not stably bind the mutated origin (chromatinized or not).

f. Supplementary Figure 4.3 looking at the stability of labelled chromatin with the bulk ORC and Cdc6 incubation can have the same comments that apply to Supplementary Figure 1.4. It is interesting to see how adding different experimental components like Cdc6 changes how well H2A dimers occupy foci and therefore how well model predicts the experimental. Do the extended incubations directly show the presence of ORC and Cdc6 affects nucleosome positioning? I would be more convinced if the incubations were tracked in the Q-trap like experiment of Figure 3.

Response: See our responses under point 16a. We looked at positioning and stoichiometry of nucleosomes in buffer (Supplementary Figure 4.1), with ORC and Cdc6 (Supplementary Figure 4.3) or with ORC, Cdc6, and MCM/Cdt-1 (Supplementary Figure 5.2), all using otherwise identical experimental conditions. We did not perform the tracking directly in the Q-trap as the extended incubation times render this experimentally intractable.

17. Medium: Figure 5 and Supplementary Figure 5s

a. In Figure 5a, the first representative cartoon of MCMs should be labelled. Horizontal opaque red blobs are not massively distinct from vertical opaque red blobs.

Response: We thank the referee for this suggestion and have included the labeling as requested.

b. Again explaining why design the reaction to happen in a tube before introducing into the flow cell and not do the reaction in the Q-trap for 30 minutes before tethering to beads would be useful? Or why not tether DNA and then introduce to protein channel in the Q-trap? It would be nice to have both experiments and/or an explanation for this chosen experimental set up.

Response: See our response under point 16a.

c. Are these experiments conducted with labelled histones? If so, make clearer, if not, why not?.

Response: See our responses under points 13 and 14.

d. Why no representative kymographs of the labelled MCMs on DNA? Examples of MCM localizing at the ARS sites with nucleosomes can be included.

Response: Regarding considerations about kymographs versus confocal scans, see our response under point 13. In the context of Figure 5, we have now included typical confocal scans, in the color channels analyzed: red for MCM and blue for H2A, along with the histograms representative of each experimental scenario below the schematics and above the histograms in Figure 5b. We have also included a new Supplementary Figure 5.3 showing representative kymographs of MCM in ARS1 origin (chromatinized or not) while imaging at one frame per minute.

e. Supplementary Figure 5.1, again why make reader do the maths themselves to interpret the data of the fraction of MCMs bound to origin bin increases with ARS and nucleosomes compared to the other conditions?.

Response: We have included the percentage of MCM bound to origin bin in Supplementary Figure 5.2.

f. Supplementary Figure 5.2 have the same comments that apply to Supplementary Figure 1.4.

Response: See our response under point 10. We have changed Supplementary Figure 5.2 as per the request of the referee.

Reviewer #3 (Remarks to the Author):

The manuscript by Sánchez and colleagues investigated how the presence of origin-flanking nucleosomes affects the binding and dynamics of ORC proteins and MCM loading on the origin recognition sequences, mainly by using the in vitro single-molecule optical tweezer coupled with fluorescence assay. They found that the presence of nucleosomes surrounding origins provides a mild increase in local binding but a reduction in mobility of ORC, and contributes to the

loading and spatial constraint of MCM. In addition, they investigated the influence of ORC and MCM on the nucleosomes around the origin, and found ORC can broaden the spatial distribution and reduce the stoichiometry of H2A in the nucleosomes and observed a protective role for MCM in the maintenance of nucleosome. By using the purified system, this study directly provides new insights into the important question of how the local nucleosomes function in the origin licensing during DNA replication in yeast, and also provides a good platform to further explore the mechanism and the epigenetic regulation of chromatin replication processes.

Overall, the experiments largely addressed the effect of origin-flanking nucleosomes on the binding and dynamics of ORC proteins and MCM.

Response: We thank the reviewer for recognizing the relevance of the questions addressed in this manuscript and for acknowledging that the data support our claims.

However, there are a few questions that should be clarified, as described in details below:

1. It would be helpful to give more details on the DNA construct used. As for the Figure 1b up right panel, the author should give more details on the center fragment of the DNA template, including to clarify the length of the ARS1, and the distance between the ARS1 and the flanking NPS.

Response: To accommodate this request, we have included a more detailed map of the fragment containing the ARS and the flanking NPSs in a new panel of the Supplementary Figure 1.2. The ACS element of the ARS1 origin of replication is separated from the NPS by 11 bp upstream of the sequence; the B2 element of the ARS1 origin of replication is separated from the NPS by 60 bp downstream of the sequence.

2. Does the distance between the ARS1 and the flanking nucleosomes affect the results, for example, the binding of ORC and MCM on ARS1? The author should try different lengths of the distance to clarify how it affects and clarify why they choose this distance in the study.

Response: This is an interesting question that we are experimentally pursuing, but we consider its answer to be beyond the scope of this paper. In the present work, chose a DNA construct that was previously been shown to be efficient for MCM loading (Miller et al., Nature 2019). Furthermore, this DNA permit to assembly nucleosomes with a separation that is compatible with the length of nucleosome-free regions described in the yeast genome (Eaton et al., Gene Dev., 2010).

3. In the Figure 1b right lower panel, why only one single spot is detected in the middle? it should see two spots for the two nucleosomes located at the two NPSs?.

Response: The two labeled nucleosomes flanking the origin are separated by 144 bp. While our localization error is about 100 bp, the resolution with which we can identify two simultaneously emitting fluorophores is lower, which effectively results in the detection of the two nucleosomes in a single diffraction-limited spot. See also our responses to Referee 1 regarding the bin size of the histograms.

4. For the Figure 2, 4 and 5, the typical confocal scans for the signal of fluorescent-labeled ORC, MCM or the nucleosomes should be presented for the four DNA and chromatin templates.

Response: We refer to our answer to Referee 2, point (13a) and point (17d). We have included typical confocal scans along with the histograms representative of each experimental scenario. We have include new supplementary Figure 3.1 and Figure 5.1 showing representative kymographs.

5. Only observed the change of fluorescence labeled-H2A cannot indicate the change of NPS site occupancy. It would be helpful to observe the change of fluorescence labeled-H3 or H4 to confirm these results.

Response: We agree with the Referee that the change in fluorescence-labeled H2A alone cannot indicate the change in NPS site occupancy. A more complete approach would be double labelling of nucleosomes (e.g. both dimers and tetramers), but this is challenging. The alternative experiments using only H3 or H4 labeling were considered to be less informative than using H2A, as they only report the presence of tetrasomes with absolute certainty.

6. We know a histone octamer wrapped more than 140 bp DNA to form a nucleosome. To unwrap one intact nucleosome, the length change of DNA should be around 45 nm, which has been shown by several force spectroscopy studies on the nucleosome unwrapping process. But in this work, the author claimed that they observed one jump at a force of 17.6 pN, with the length increments of 24.8 nm for the nucleosome force spectroscopy. It looks the nucleosomes used in this study were not well reconstituted, with only sub-nucleosomes or tetrasomes are formed.

Response: The Referee is correct that the unwinding of a complete nucleosome releases about 45 nm. However, what we show here, in agreement with recent studies using force spectroscopy (Diaz-Celis et al., PNAS, 2022), is the length increase **detected** under our experimental conditions.

We note additionally that our fluorescence data are in quantitative agreement with our force data. We show by force spectroscopy (Supplementary Figure 1.5c) that 88.5% of the molecules have two nucleosomes (or at least hexasomes). From the fluorescence data, we deduce that the probabilities of having a full occupancy of the NPSs ($p_{\text{occupancy}} = 1$) and having a dimer bound to a tetrasome ($p_{\text{h2a}} = 0.77$), see Figure 1, Supplementary Figure 1.3, and the description of the probability model in the Methods section). Based on these values, we calculate the fraction of foci with two or more labeled H2A to be 96%, in excellent agreement with the corresponding value of 88.5% deduced from force spectroscopy. This information has been added to the main text.

7. It should be interesting to observe, how the binding of ORC or MCM affect the property, including the stability or unfolding kinetics of the flanking nucleosomes?

Response: This is an interesting question but we consider its answer to be beyond the scope of this paper.

Additional modifications to the manuscript:

- a. We have clarified which proteins used in this study are labelled by adding a superscript with the name of the fluorophore to the name of the protein or protein complex used throughout the experimental portions of the manuscript.
- b. We have indicated in the captions of Figure 1, Figure 4, Figure 5, Supplementary Figure 1.4, Supplementary Figure 4.1, Supplementary Figure 4.2, Supplementary Figure 4.3, Supplementary Figure 5.1, and Supplementary Figure 5.2 that the spatial distributions of the foci are estimated immediately after introduction into the flow cell.
- c. We have indicated in the captions of Figure 2 and Supplementary Figure 2 that the spatial distributions of the foci are estimated immediately after introduction immediately after incubation in Ch4 and first illumination in Ch3 of the flow cell.
- d. We have clarified the title of the captions of Figure 2, Figure 3, Figure 4, Supplementary Figure 2, Supplementary Figure 3.2, Supplementary Figure 4.2, Supplementary Figure 4.3, Supplementary Figure 5.1, and Supplementary Figure 5.2 to indicate the state of the DNA (containing an ARS1 (or mutated) origin (chromatinized or not)).
- e. We have clarified the schematics of Figure 2, Figure 4, Supplementary Figure 2, Supplementary Figure 3.2, Supplementary Figure 4.1, Supplementary Figure 4.2, Supplementary Figure 4.3, Supplementary Figure 5.1, and Supplementary Figure 5.2 by indicating the scan acquisition frequency.

References cited in this response:

- Li NN, et al. Structure of the origin recognition complex bound to DNA replication origin. Nature 559, 217-+ (2018).
- Remus D, Beuron F, Tolun G, Griffith JD, Morris EP, Diffley JFX. Concerted Loading of Mcm2-7 Double Hexamers around DNA during DNA Replication Origin Licensing. Cell 139, 719-730 (2009).
- Miller TCR, Locke J, Greiwe JF, Diffley JFX, Costa A. Mechanism of head-to-head MCM double-hexamer formation revealed by cryo-EM. Nature 575, 704-+ (2019).
- Sanchez H, et al. DNA replication origins retain mobile licensing proteins. Nature Communications 12, (2021).

- Eaton ML, Galani K, Kang S, Bell SP, MacAlpine DM. Conserved nucleosome positioning defines replication origins. *Gene Dev* 24, 748-753 (2010).
- Diaz-Celis C, et al. Assignment of structural transitions during mechanical unwrapping of nucleosomes and their disassembly products. *P Natl Acad Sci USA* 119, (2022).

REVIEWERS' COMMENTS

Reviewer #1 (Remarks to the Author):

The modification made to the manuscript are clear and have improved it. However, the authors didn't really address my last questions.

-And is it possible to reduce the bin size in order to get more spatial resolution?

It seems that they choose very conservative and with that they lose information. I would encourage them to push the limits a bit more.

Reviewer #2 (Remarks to the Author):

The authors addressed all of my comments. In my opinion, the manuscript is ready for publication in Nature Communications.

Reviewer #3 (Remarks to the Author):

The revised manuscript by Sánchez and colleagues has been considerably improved and contains important additional data. The authors replied to the suggestions and have mostly addressed my main comments with further explanations.

However, for the H3 or H4 labeling, they argue that it is less informative than using H2A labeling, which I do not agree with. The H2A/H2B dimer are more dynamic than H3/H4 in nucleosome, and missing of H2A cannot indicate well the change in NPS occupancy.

Nevertheless, I leave this revision up to the authors and have no further comments preventing publication.

Delft, October 11, 2023

We thank again the reviewers of our manuscript for their interest in the work and positive endorsement. Below, we provide a point-by-point response (in blue) to their comments (in black italics).

Kind regards,

The authors

REVIEWER COMMENTS

Reviewer #1 (Remarks to the Author):

The modification made to the manuscript are clear and have improved it. However, the authors didn't really address my last questions.

-And is it possible to reduce the bin size in order to get more spatial resolution?

It seems that they choose very conservative and with that they lose information. I would encourage them to push the limits a bit more.

Response: We believe that the conservative bin size chosen to display the data adequately reflects our best estimate of the absolute localization of the proteins studied (without risk of overinterpretation) and supports the conclusion drawn in this manuscript. We are reluctant to switch to relative localizations (which we know more accurately), because it is of interest in our measurements to know the exact positions of ORC and MCM. Only in this way can we compare them to the expected locations of nucleosomes. Furthermore, in order to show display an enhanced relative localization accuracy, we would decrease the bin size, but this would also reduce the statistics per bin. Given that these are not obvious improvements that would change the interpretation of results or benefit readers, we have left the bin size unchanged.

Reviewer #2 (Remarks to the Author):

The authors addressed all of my comments. In my opinion, the manuscript is ready for publication in Nature Communications.

Response: We thank the reviewer for the positive feedback.

Reviewer #3 (Remarks to the Author):

The revised manuscript by Sánchez and colleagues has been considerably improved and contains important additional data. The authors replied to the suggestions and have mostly addressed my main comments with further explanations.

However, for the H3 or H4 labeling, they argue that it is less informative than using H2A labeling, which I do not agree with. The H2A/H2B dimer are more dynamic than H3/H4 in nucleosome, and missing of H2A cannot indicate well the change in NPS occupancy.

Nevertheless, I leave this revision up to the authors and have no further comments preventing publication.

Response: We thank the reviewer for the positive feedback. Regarding the information that can be acquired from histone labeling, we note that imaging fluorescently labeled H2A allows us to extract the occupancy probability for an NPS site together with the probability of forming a complete histone octamer (see Methods and Supplementary Figure 1.3). However, with H3/H4 labeling only, one can only determine the occupancy probability for an NPS site. This is therefore less informative.